# Divergent sensory investment mirrors potential speciation via niche partitioning across *Drosophila*

Ian W Keesey*, Veit Grabe, Markus Knaden[†]*, Bill S Hansson[†]*

Max Planck Institute for Chemical Ecology (MPICE), Department of Evolutionary Neuroethology, Jena, Germany

**Abstract** The examination of phylogenetic and phenotypic characteristics of the nervous system, such as behavior and neuroanatomy, can be utilized as a means to assess speciation. Recent studies have proposed a fundamental tradeoff between two sensory organs, the eye and the antenna. However, the identification of ecological mechanisms for this observed tradeoff have not been firmly established. Our current study examines several monophyletic species within the *obscura* group, and asserts that despite their close relatedness and overlapping ecology, they deviate strongly in both visual and olfactory investment. We contend that both courtship and microhabitat preferences support the observed inverse variation in these sensory traits. Here, this variation in visual and olfactory investment seems to provide relaxed competition, a process by which similar species can use a shared environment differently and in ways that help them coexist. Moreover, that behavioral separation according to light gradients occurs first, and subsequently, courtship deviations arise.

**\*For correspondence:**
ikeesey@ice.mpg.de (IWK);
mknaden@ice.mpg.de (MK);
hansson@ice.mpg.de (BSH)

[†]These authors contributed equally to this work

**Competing interests:** The authors declare that no competing interests exist.

## Introduction

The genus *Drosophila* provides an incredible array of phenotypic, evolutionary and ecological diversity (*Jezovit et al., 2017*; *Keesey et al., 2019*; *Keesey IW et al., 2019*; *Markow, 2015*; *Markow and O'Grady, 2007*; *O'Grady and DeSalle, 2018*). Members of this genus, which provides roughly 1500 species including the model organism *D. melanogaster*, inhabit all continents except Antarctica, and occur in almost every type of environment. Due to their vast variation in behavioral, morphological and natural history traits, the comparison of vinegar flies provides an enormous potential for the understanding of driving forces in evolutionary processes. In particular, the feeding, courtship and breeding sites of this genus are tremendously diverse, including both generalists and specialists, and spanning extreme dietary variation and host utilization such as different stages of fruit decay, as well as flowers, mushrooms, sap or slime flux, rotting leaves, cacti and many other sources of microbial fermentation. It is important to note that preferences in feeding and oviposition have shifted numerous times, and closely related species are known to utilize different types of food resources (*Crowley-Gall et al., 2019*; *Crowley-Gall et al., 2016*), or to visit a host at different stages of decay (*Karageorgi et al., 2017*; *Keesey et al., 2015*; *Ometto et al., 2013*). At the same time, it is common to find phylogenetically distant species using the same host, or living in overlapping environments (*Hey and Houle, 1987*; *Lachance et al., 1995*; *Martin, 1998*; *Taylor, 1987*; *Taylor and Powell, 1978*). Therefore, the spatial distribution of species over discrete patches of an ecosystem, such as within a temperate forest, might vary according to discrete microhabitats. While little ecological information is available for a majority of the non-*melanogaster* species, it has been shown repeatedly that many of the members of the *obscura* species group overlap geographically as well as ecologically in their utilization of temperate forest ecosystems (*Bächli et al., 2006*; *Burla et al., 1986*; *Michell and Epling, 1951*). Moreover, these local environments may create different selective

and energetic pressures for neuroanatomy (*Niven and Laughlin, 2008*) that in turn could provide an opportunity and rationale for the coexistence of many species within a single habitat or ecological niche (*Finke and Snyder, 2008*; *Griffin and Silliman, 2011*).

In an earlier paper, we showed that robust idiosyncrasies exist between visual and olfactory investment across this genus (*Keesey et al., 2019*), including many examples of inverse variation within a subgroup, and between sympatric species and subspecies that utilize seemingly identical host plants or food resources. However, most vinegar fly species have an understudied ecology, and other than information about where and when they were collected for laboratory establishment, we often know very little about their natural habitats or ecological preferences. The aim of the present paper is to determine whether behavioral, phenotypic, and neuronal differences between close relatives all combine to support the coexistence of different species within a single ecological habitat. As predicted by our initial hypotheses, we document that these sensory traits vary significantly between two close relatives within the *obscura* group – *D. subobscura* and *D. pseudoobscura* – and we examine in detail the potential driving forces of speciation, such as biotic and abiotic factors, including courtship modalities and phototactic response. Next, we expand our research objectives to predict sensory variation in monophyletic species based on our hypothesis from the *obscura* group, including *D. persimilis*, *D. affinis* and *D. bifasciata*, where we test our hypothesis that this sensory variation will consistently occur across subgroups within the *obscura* clade. Here, we assert that even between close, phylogenetic relatives as well as sympatric species, these differences in visual and olfactory sensory investment are strongly apparent. We propose that these sensory differences could reduce interspecies competition via resource partitioning and through innate variation in microhabitat or microclimate preferences, thus promoting speciation, novel niche establishment, as well as stabilizing selection using natural sensory trait variation across this important genus of insects. Moreover, behavioral differences related to phototaxis appear to be more significant between sympatric species, and thus niche partitioning may be the initial driving force, whereas differences in courtship then promote and maintain speciation events.

## Results

### External morphology of sensory systems

In order to examine the sensory traits of five closely related and often co-occurring species – *D. persimilis*, *D. affinis*, *D. bifasciata*, *D. subobscura* and *D. pseudoobscura* – we quantified their visual and olfactory investment by first measuring the external morphology of their visual and olfactory systems. Here, we first focused on the two best studied members, *D. subobscura* and *D. pseudoobscura*, where previous work has already suggested potential differences (*Keesey IW et al., 2019*; *Ramaekers et al., 2019*; *Tanaka et al., 2017*). Eight to 10 females of these species were photographed using a Zeiss AXIO microscope, including lateral, dorsal, and frontal views. We then measured across a variety of physical characteristics, such as surface areas of the compound eye, antenna, maxillary palps, ocelli, and overall body size, as well as head, thorax, abdomen and femur length. We also generated metrics for the number of ommatidia as well as measures of trichoid sensilla for each species. It is not possible to distinguish between antennal trichoid one (at1) and antennal trichoid four (at4), at least not based on morphology alone. Therefore, we clarify herein that trichoid measurements refer collectively to both at1 and at4 across those examined *Drosophila* species. In general, we found that *D. subobscura* possessed much larger eyes in regards to surface area, as well as 25–30% more ommatidia than its close relative, *D. pseudoobscura*, though ommatidia diameter was identical (*Figure 1A–E*; *Figure 1—figure supplement 2F*). While there was some variation in individual size within and between species (with *D. subobscura* exhibiting larger dimensions in all measured body parts; *Figure 1—figure supplement 1A–E*), we note that eye surface area was consistently correlated with ommatidium number (*Figure 1D*), suggesting that eye surface area provides a good approximation of visual investment. Here, we note that both of these two species had a nearly identical linear relationship between surface area and number of ommatidia (*Figure 1D*), with *D. subobscura* possessing larger eyes. While *D. pseudoobscura* possessed smaller eyes and a reduced ommatidium count, females of this species instead displayed larger antennal surface areas relative to *D. subobscura* females (*Figure 1A–C,F*). Interestingly, not all metrics related to sensory organs on the head were different between these closely related species. For example, the maxillary

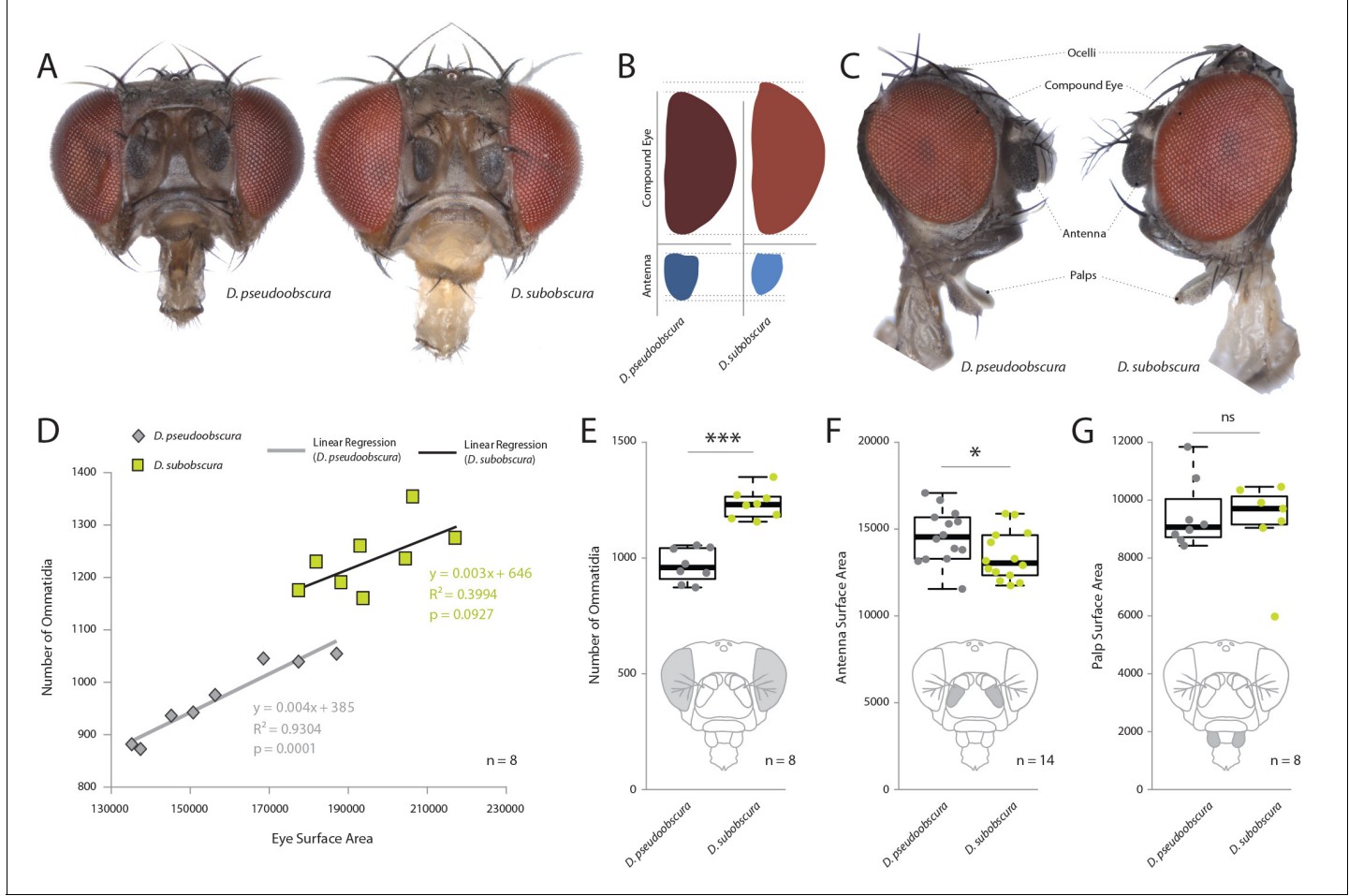

**Figure 1.** Comparative morphology of external sensory systems. (**A**) Examples of frontal head replicates, where eye and antenna surface area was measured from females. Note the differences in pigmentation, as well as the size of the compound eye and third antennal segment from both species. (**B**) Side-by-side comparison of both the compound eye (red) and third antennal segment (blue) from each fly. (**C**) Example of lateral views used for measurements, including compound eye surface area, antennal surface area, and maxillary palp surface area. (**D**) Intra- and interspecies correlations between eye surface area and the number of ommatidia for *D. subobscura* (yellow) and *D. pseudoobscura* (grey). (**E–G**) Comparison of ommatidia counts (**E**), antennal surface (**F**), and palp surface (**G**) for both species. Boxplots represent the median (bold black line), quartiles (boxes), as well as 1.5 times the inter quartile range (whiskers). Mann-whitney U test; ***, p<0.001; *, p<0.05; ns, p>0.05.

The online version of this article includes the following source data and figure supplement(s) for figure 1:

**Source data 1.** Morphometrics across the obscura group.
**Figure supplement 1.** External morphometrics of two *obscura* species.
**Figure supplement 2.** Additional metrics from sensory systems.
**Figure supplement 3.** Measurements of ommatidia diameter.

palps (*Figure 1G*) did not display any significant variation in surface area, but we do note differences in the ocelli (*Figure 1—figure supplement 2C–E*). Thus, these changes to sensory systems on the head appear mostly restricted to the antenna and to the visual sensory modalities.

## Comparative neuroanatomy of visual and olfactory investment

As we had already established divergent external morphology between these two species, especially in regards to vision and olfaction, we next focused our attention on the primary processing centers in the brain for these sensory systems, including the antennal lobe (AL) and optic lobe (OL) (*Figure 2AB*). After correcting for adult size (using the remaining hemisphere or central brain volume as a reference for each species; in grey) (*Keesey et al., 2019*), we identified a relative increase of the AL size for *D. pseudoobscura* (*Figure 2C*), as well as a relative decrease of the size of its OL

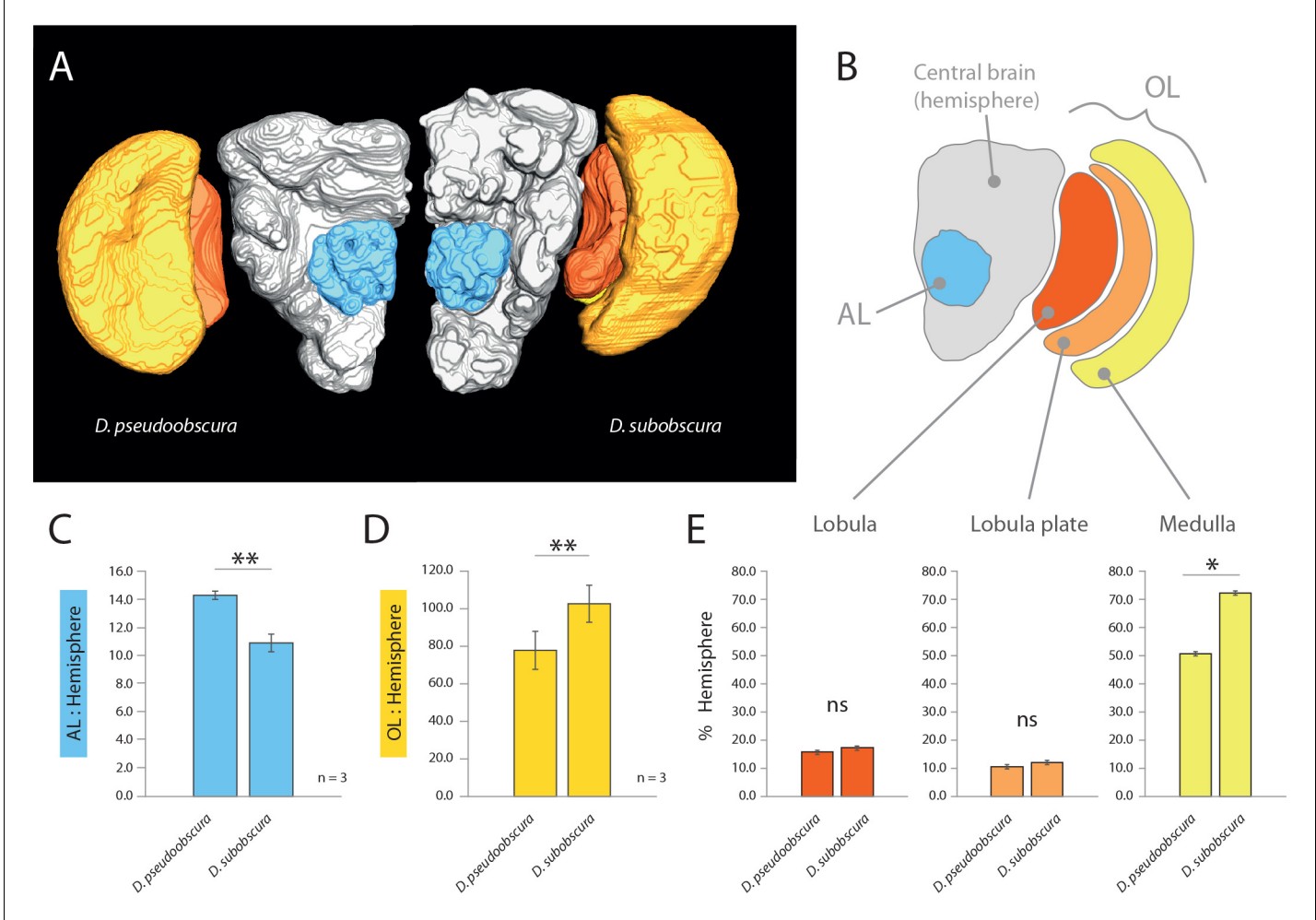

**Figure 2.** Comparative morphology of primary processing centers in the brain. (**A**) Three-dimensional reconstructions of the neuropils of *D. pseudoobscura* and *D. subobscura* adult females. (**B**) Diagrammatic representation of the brain, with color-coded and labeled volumetric sources. AL, blue; hemisphere, grey; optic lobe, OL, with medulla (yellow), lobula (red) and lobula plate (orange). (**C–E**) Relative size of AL (**C**), OL (**D**), and lobula plate, lobula, and medulla (**E**) as compared to respective hemisphere [%].

The online version of this article includes the following source data for figure 2:

**Source data 1.** Metrics associated with visual and olfactory neuropils.

(*Figure 2D*) when compared to the same neuropils for *D. subobscura* adults. These inverse values between the two sensory systems correspond strongly to the variations we measured in the external morphology, where one species had larger eyes but smaller antennae, and vice versa. Moreover, to highlight the regions of the OL that show the largest increases, we provide similar metrics for relative size for the lobula plate, lobula and the medulla (*Figure 2E*), where all brain regions (again when corrected for total brain size; in grey) are bigger in *D. subobscura*, but only the medulla is significantly larger.

## Courtship and mating behavior differences between *obscura* species

In order to ascertain the possible ramifications of inverse eye and antenna variation between our two species, we proceeded to examine behaviors related to mate selection and courtship. Previous research has shown that *D. subobscura* displays light-dependent courtship, and will not successfully copulate in the dark (*Wallace and Dobzhansky, 1946*; *Grossfield, 1971*; *Spieth, 1952*). Counter to this, *D. pseudoobscura* mating is light-independent, and courtship can successfully occur regardless of light conditions (*Brown, 1964*; *Wallace and Dobzhansky, 1946*; *Grossfield, 1971*; *Ripfel and*

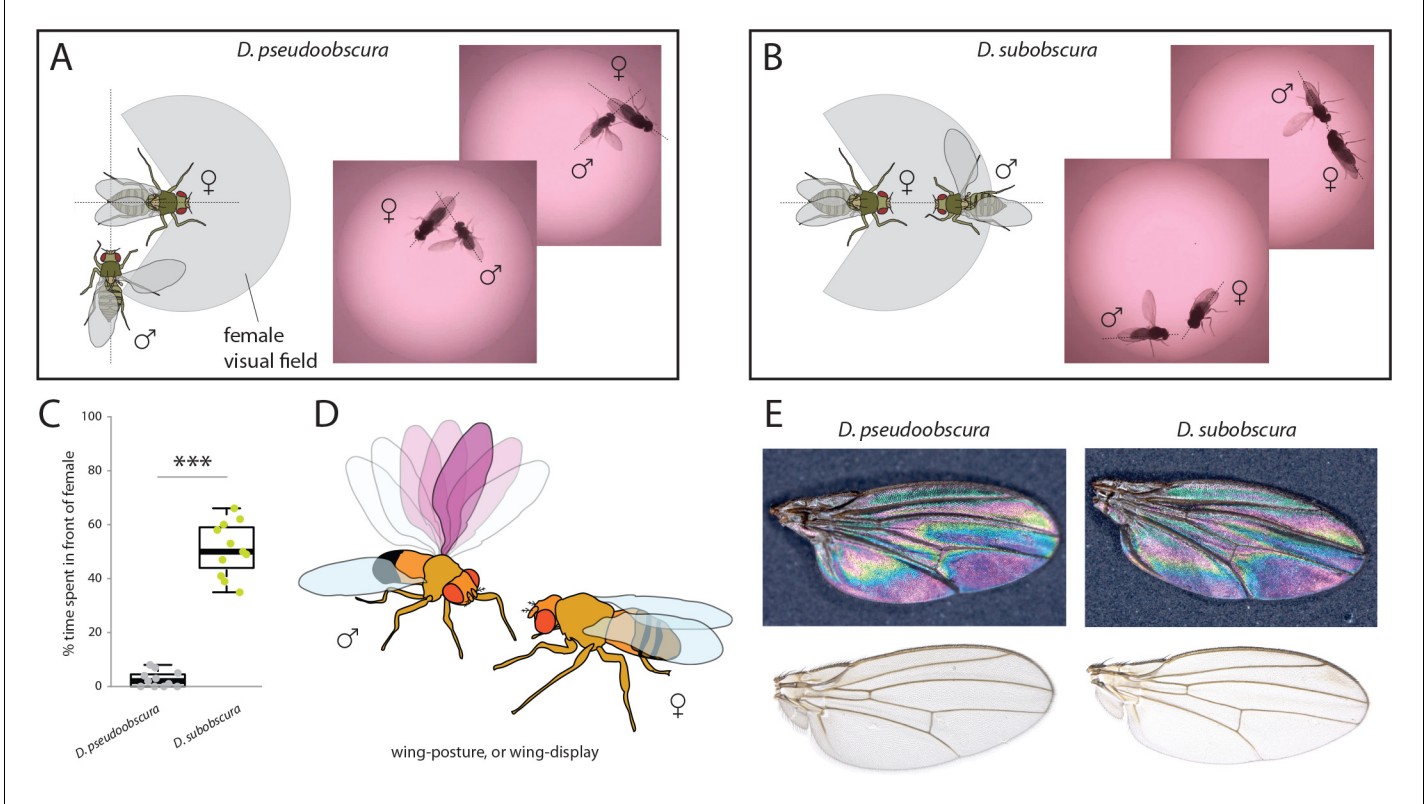

**Figure 3.** Courtship and mating behavior. (A–B) Images of courtship for *D. pseudoobscura* (A) and *D. subobscura* (B) and schematic of the behavior, whether the male is in- or outside the predicted visual field of the female. (C) Time that males spent within the female visual field during courtship. Boxplots represent the median (bold black line), quartiles (boxes), as well as 1.5 times the inter quartile range (whiskers). Mann-Whitney U test; ***, p<0.001 (D) Diagram of *D. subobscura* wing display by the male, where no wing vibration was observed, and instead, a discrete range of wing angles was presented and maintained towards the female mating partner during courtship. (E) Stable structural wing interference patterns observed across the otherwise clear wings of males of both species.

*Becker, 1982*; *Spassky, 1967*). Therefore, as we wanted to observe and dissect the behavioral motifs and succession of events that lead to successful courtship, we performed courtship trials under identical conditions for both species. We recorded videos (*Video 1* and *Video 2*) using virgin males and females that were introduced into a small courtship arena (*Figure 3*). Several differences were immediately noted between the species. *D. pseudoobscura* males oriented themselves either behind or to the side of the female during courtship, often forming a right angle to her with the male head focusing on the last few abdominal segments of the potential mate (*Figure 3A*). Next,

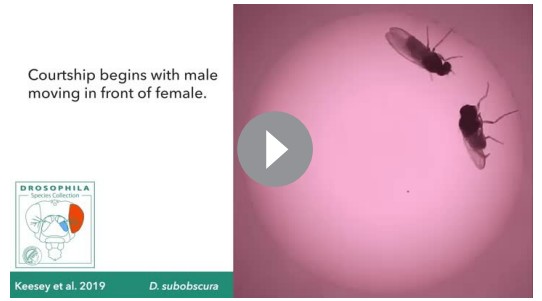

**Video 1.** Courtship behavior video clip examples for *D. subobscura* adults.
https://elifesciences.org/articles/57008#video1

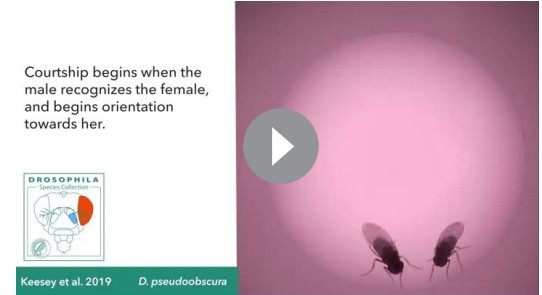

**Video 2.** Courtship behavior video clip examples for *D. pseudoobscura* adults.
https://elifesciences.org/articles/57008#video2

this species performed characteristic wing vibrations and singing, with the outstretched wing always nearest to and in the direction of the head of the female (*Figure 3A*), and with the male constantly in pursuit of the female from behind or from the side. It is not clear if larger pedicel size (i.e. Johnston's organ) correlates with species that perform songs, but future work will address this hypothesis. In stark contrast, observations of the courtship of *D. subobscura* showed that the males of this species often dart around in a circular arc to put themselves directly in front of the path of the female, and appear to arrest her movement (*Figure 3B,C*). This frontal positioning by the *D. subobscura* male results in most of the subsequent courtship behaviors occurring in front of the female and within her visual field, including the male wing displays. Here, *D. subobscura* was not observed to vibrate the outstretched wing (unlike *D. pseudoobscura* males, which are known to sing), and instead, seemed to angle or tilt the outstretched wing during the display, possibly as a flash of color via wing interference patterns (WIPs) (*Shevtsova et al., 2011*) or another visual exhibition for the female (*Figure 3B,D,E*).

## Phototactic responses by close-relatives of the *obscura* group

Given that we had established that differences in compound eye and antenna sizes correlated with differences in courtship behavior, we next examined whether the morphological investments played any additional role in ecological decisions related to environmental preferences. Here we utilized a simple Y-tube two-choice behavioral assay, where adult flies from each species could select between a light or dark environment (*Figure 4A*). We observed that the smaller-eyed *D. pseudoobscura* significantly preferred to enter the Y-tube arm that was in shadow and darkened (*Figure 4B*). In contrast, the larger-eyed adult *D. subobscura* significantly preferred the Y-tube arm that was in full light.

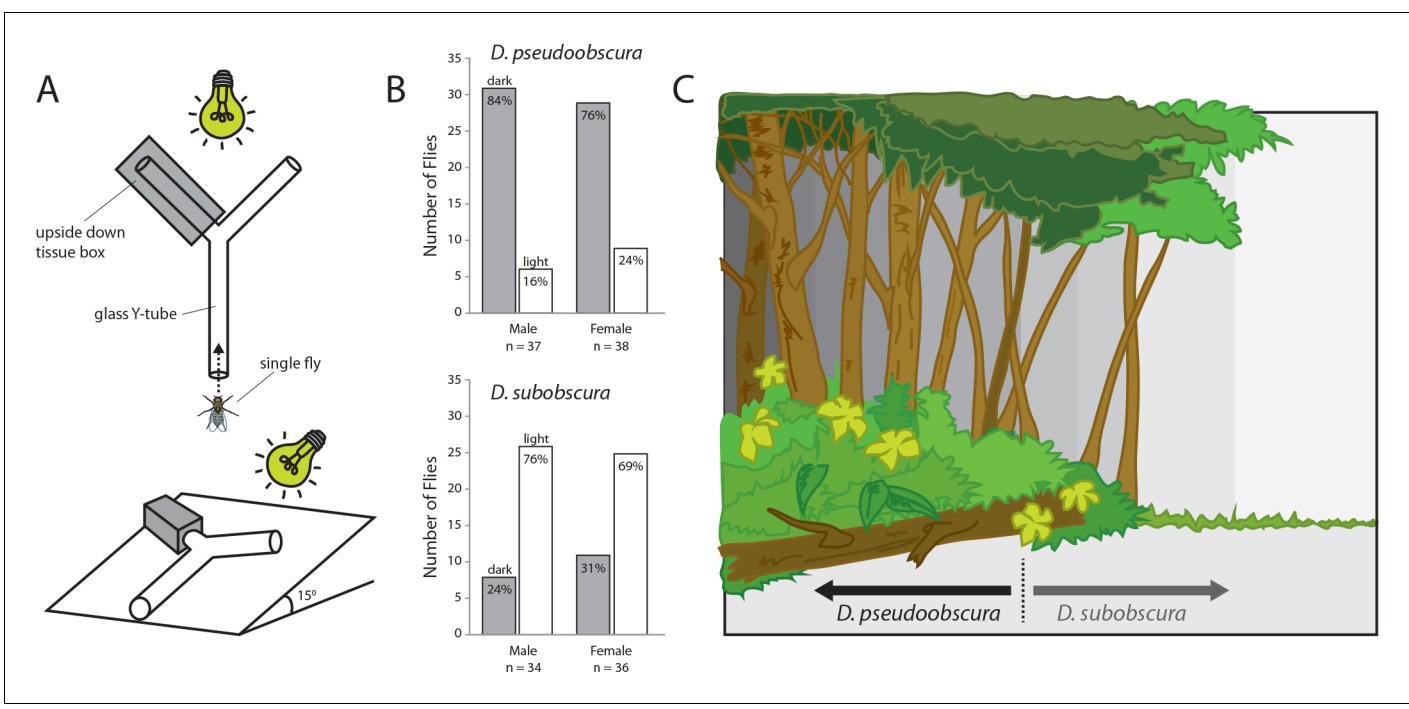

**Figure 4.** Light preferences and hypothesized niche partitioning by both species. (**A**) Two diagram views of the Y-tube phototactic response paradigm. Single flies were allowed to choose between either a well-lit or a darkened arm of a Y-tube. (**B**) Percentages of male and female flies of both species choosing the well-lit or darkened arm of the y-tube. (**C**) Diagram of ecological niche partitioning where our closely-related *Drosophila* species divide spatially across microhabitats within the same environment, and where light gradients act as an isolation barrier. Here we propose that these *obscura* species, despite sharing a forest ecology, create a reduction in either host resource or mating competition via their different preferences toward edge and open canopy environmental conditions, as related directly to their innate preferences for light intensity.

The online version of this article includes the following source data for figure 4:

**Source data 1.** Y-tube assay and phototaxis dataset.

## Expansion of hypotheses to include additional species

Using the same behavioral, phenotypic and morphological examinations, we tested our hypotheses of sensory trait variation across three additional members of the *obscura* group. Here we included *D. persimilis*, which is a well-studied, sympatric species for direct comparison to *D. pseudoobscura*, as well as *D. affinis*, which also shares North American habitats with these two species. We also added *D. bifasciata*, which is a member of the *obscura* subgroup, and is endemic to Asia. In total, these five vinegar flies represent a reasonable phylogenetic spectrum, and provide example species from four of the main subgroups of the *obscura* clade. After collecting images from several angles including frontal views (*Figure 5A*), we analyzed in depth the visual and olfactory morphology of each new species, ultimately generating an eye-to-funiculus ratio (EF ratio) for each fly (*Figure 5A– C*), which has been used previously to compare sensory systems from species of differing absolute size (*Keesey IW et al., 2019*). Moreover, we examined three populations of each species in order to examine the consistency of EF Ratio within and between our five *obscura* species. Subsequently, we again repeated the same behavioral regimes using these new *obscura* members, including both y-tube phototaxis as well as species-specific courtship ethology. In these cases, we documented a rather steady variance in positive or negative phototaxis across this growing phylogenetic examination, with our initial two species (e.g. *D. pseudoobscura* and *D. subobscura*) representing the two extremes (*Figure 5D*). Similarly, we also observed a consistent change in male courtship behavior, as measured by the percentage of time the male of each new species spent in front of the female during his dance or mating display. By documenting the relationship between our ommatidium counts and our estimates of the surface area of the compound eye, we conclude that diameter of ommatidia does not vary between our species (*Figure 1—figure supplement 3*), and we conclude that surface area is a consistently accurate metric for estimating the number of visual facets in the *obscura* group (*Figure 5E*). We do note variation in the absolute size of flies within each species, and future research should examine this aspect for additional assessments of sensory plasticity or constraints, perhaps related to population density. However, to generate similar sized adults, we controlled rearing density for consistent adult sizes across both morphological and behavioral assessments. Interestingly, we also describe correlations between EF ratio and phototaxis across our species, as well as the percentage of courtship the male spends in front of the female (*Figure 5F,G*). In both instances, larger EF ratios correspond tightly with increases in positive phototaxis (i.e. attraction to light) and correspond strongly with increases in courtship behaviors generated while in front of the female, which we presume are related to the importance of visual sensory signals. Here we note that some of the largest behavioral differences (i.e. slope between individual species) still occur between closest phylogenetic relatives, including *D. pseudoobscura* and *D. persimilis* (*Figure 5F,G*), which represent the most-studied and well-published sympatric species pair from the *obscura* group. In these sympatric species, we see larger behavioral variations (y-axis) than changes in morphology (x-axis), suggesting that perhaps even small tradeoffs in olfactory or visual sensory systems can generate robust changes in behavior (*Figure 5F,G*). Here, we note that changes in phototaxis between sympatric species appear to be stronger and more acute than changes in courtship dynamics (*Figure 5D,F,G*). Moreover, we observe that the slope of the correlation between phototaxis and EF Ratio for the sympatric species is greater than the slope related to the rest of the *obscura* subgroup (*Figure 5F*; orange vs grey).

## Discussion

When we imagine examples of isolation barriers, we often consider those that are distinctly physical in nature, such as a mountain range or a remote island biogeography. However, sensory isolation barriers also exist, including differences in pheromone chemistry between geographically overlapping species (*Chung et al., 2014*; *Löfstedt, 1993*; *Löfstedt et al., 1991*; *Mitchell et al., 2015*), or variations in the songs and auditory repertoires of crickets, frogs and birds (*Blair, 1974*; *Hobel and Gerhardt, 2003*; *Kirschel et al., 2009*; *Walker, 1974*). In this study, we hypothesize that sources of light gradients may also create strong selective pressures and isolation mechanisms that in turn lead to speciation events or stabilizing selection for opposing phototaxis within otherwise overlapping habitats, such as arboreal forests. Moreover, we propose that these light gradients most likely work in tandem with inverse changes in pheromone or chemosensory changes to further provide avenues for species divergence. Arboreal forest microhabitats have been addressed previously as sources for

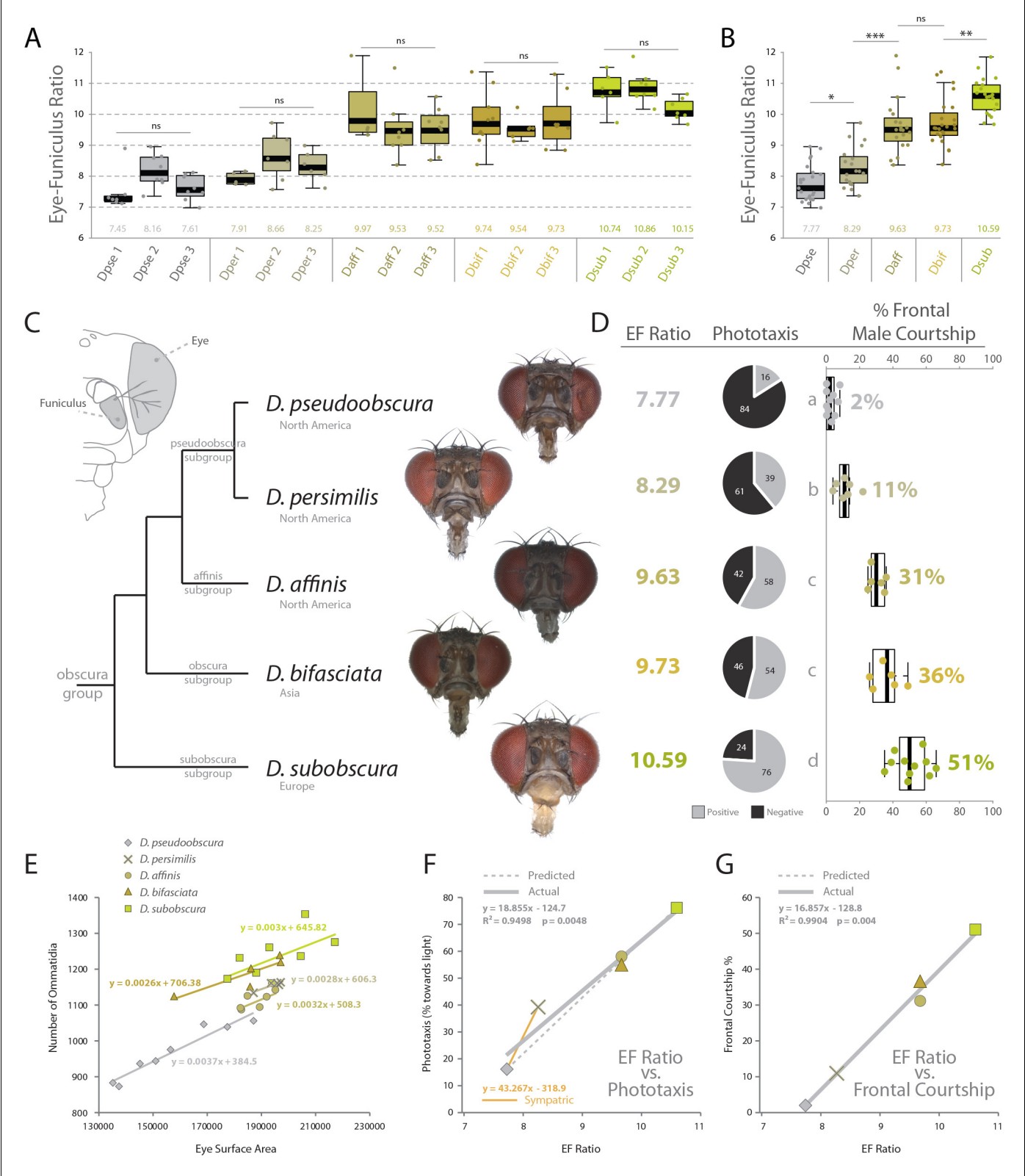

**Figure 5.** Behavioral relevance of sensory investment in 5 species of the *obscura* group. (**A**) Measures of eye-to-funiculus ratio (EF ratio) across 15 populations of *obscura* species. (**B**) Statistical assessments of average EF Ratio between each species. (**C**) Diagram of morphology used to generate EF ratio for each species and the phylogenetic relationship of these five members of the *obscura* group (***Crysnanto and Obbard, 2019***; ***O'Grady, 1999***), as well as examples of the frontal head images used to collect certain morphological data. Additional images are available with the online version of *Figure 5 continued on next page*

*Figure 5 continued*

this publication. (D) Eye-to-funiculus ratio for each measured species, as well as the male phototactic response during y-tube trials. More details about phototaxis behavioral regimes are available in *Figure 4*. In addition, courtship videos from each species were addressed to quantify the amount of time the male spent in courtship, and shown are what percentage of that courtship time was spent in front of the female (see *Figure 3* and supplementary videos for more detail). (E) Morphological measurements of eye surface area and the number of ommatidia collected from lateral views of each species. (F) Correlation between EF ratio and positive phototaxis for all tested *obscura* species. Our hypothesized correlation is shown with a dashed line, which stems from initial comparison of just two species (*D. pseudoobscura* and *D. subobscura*), while the actual correlation following the additional analyses of three new species is shown using a solid line. (G) Correlation between EF ratio and the percentage of male courtship spent in front of the female during his dance or display ethology.

The online version of this article includes the following source data for figure 5:

**Source data 1.** Morphometrics for intraspecies and interspecies comparisons.

spatial separation between species (*Atkinson and Miller, 1980*; *Montgomery and Merrill, 2017*; *Penariol and Madi-Ravazzi, 2013*; *Taylor and Powell, 1978*), including studies directly related to the field-sampling of members of the family Drosophilidae, often with the division of species occurring in proximity to the forest edge. While the evolutionary selective pressures and their effects on the relative size of various components of the nervous system have not been previously examined, it has been suggested that sources of light may be one of the ambient forces driving the observed tradeoff in the evolution of these two sensory structures (*Keesey et al., 2019*). However, additional field studies are still needed to confirm whether these sensory investments differ outside the laboratory, and to examine how insect species sort in the wild, for example, via niche partitioning or character displacement.

Here we demonstrate that several monophyletic species within the *obscura* group, despite being close relatives, deviate significantly in regards to both eye and antenna morphology (*Figure 1*), as well as in their corresponding neuropils for vision and olfaction (*Figure 2*). In addition, we observe that this variation in sensory systems positively correlates with both courtship behavior (*Figure 3*) and environmental habitat preferences (*Figure 4*), especially as related to the relative importance of visual stimuli or sources of light, which appears to be of opposing value between these sibling species (*Figure 5*). Previous work has documented this tradeoff or inverse resource allocation between vision and olfaction across more than 60 species within the *Drosophila* genus (*Keesey IW et al., 2019*), but the ecological mechanisms and selective pressures underlying this divergence have not been studied as explicitly in monophyletic species groups or subgroups. While little ecological information is available for a majority of the non-*melanogaster* species, it has been shown repeatedly that many of the members of the *obscura* species group overlap geographically as well as ecologically in their utilization of temperate forest ecosystems (*Bächli et al., 2006*; *Burla et al., 1986*; *Michell and Epling, 1951*). In addition, there has been documentation of a recent geographical overlap between *D.pseudoobscura* and *D. subobscura* in North America, which might make for an ideal field study in the future to test our hypotheses regarding environmental partitioning and the evolution of sensory variation (*Noor, 1998*; *Pascual et al., 1998*). However, the species *D. pseudoobscura* and *D persimilis* have been well established as sympatric (*Crysnanto and Obbard, 2019*; *O'Grady, 1999*), and already act as models for genetic variation in phototaxis and courtship (*Brown, 1965*; *Brown, 1964*; *Hernández and Fabre, 2016*; *Noor and Aquadro, 1998*; *Ripfel and Becker, 1982*). However, these behaviors have not been previously studied concurrently, nor with the overlay of morphological and neurobiological data across both visual and olfactory sensory systems. In these two evolutionarily sympatric species (i.e. *D. pseudoobscura* and *D persimilis*), we demonstrate significant deviation in visual and olfactory behavior, including both phototactic response and potential visual bias during male courtship (*Figure 5D–G*), thus supporting our initial proposal that variance in sensory investment occurs even within closest, sympatric species across this genus. While we have sampled several species, it would also be important to address additional *Drosophila* species within the *obscura* group, such as *D. tristis*, *D. tsukabaensis*, *D. obscura*, *D. miranda*, *D. iowei*, *D. ambigua* and *D. helvetica*. This is especially true in cases where these species potentially share ecological overlap in habitat utilization or geography, and where genomic information is perhaps readily available for additional analyses of the molecular mechanisms for this sensory tradeoff (*Ramaekers et al., 2019*). In the present study, we test the hypothesis that close insect relatives may divide host or habitat resources through niche partitioning by inversely prioritizing the relative

importance of visual stimuli as compared to those stimuli that are olfactory. This explanation would be consistent with previous observations that monophyletic species often possess inversely correlated eye and antenna sizes despite being close relatives and despite sharing seemingly identical hosts and environmental preferences (*Gaspar et al., 2020*; *Keesey et al., 2019*; *Özer and Carle, 2020*; *Ramaekers et al., 2019*). In addition, this hypothesis continues to be consistent with the more in depth analyses afforded by the present study, which documents this sensory inversion across the *obscura* group, including across close, sympatric relatives, as well as across several populations or strains of each species (*Figure 5A,B*). Please note that surface areas of the various head and thorax, as well as overall body size, were examined for only one strain per species. It thus remains to be confirmed that the observed differences are truly interspecific. Future research should continue to address both interspecies and intraspecies variation in sensory investment, especially as related to both causal agents and genetic underpinnings of this divergence.

Microhabitats often arise in nature, as landscapes are inherently non-uniform (*Martin, 1998*; *Scheffers et al., 2014*). These ecological subdivisions have been examined in regards to cline variation or altitude (*Griffiths et al., 2005*; *Michell and Epling, 1951*; *Parsons, 1991*), as well as temperature gradients or differences in water availability (*Enjin et al., 2016*; *Parsons, 1991*; *Scheffers et al., 2014*; *Toda, 1992*). In addition, several studies have addressed microhabitat variation and its effects on species richness or biodiversity. Moreover, that plant hosts and other nutritional resources such as fungi and yeasts can differ greatly between forest edge and forest interior (*Bächli et al., 2006*; *Łuczaj and Sadowska, 1997*; *Penariol and Madi-Ravazzi, 2013*; *Toda, 1992*). Thus, it is well recognized that flora and fauna can vary in both their relative abundance as well as their innate preferences across microclimates within a single habitat, where the fitness of a *Drosophila* species is intimately tied to its ability to compete for resources within its own environmental niche (*Koerte et al., 2020*; *Qiao et al., 2019*). However, the mechanisms by which these innate animal preferences for microhabitats can generate evolutionary pressures or speciation events has not been as thoroughly documented, least of all in a set of model organisms where robust molecular genetic toolkits are available, such as those afforded by the *Drosophila* genus.

Other insects have been examined for their differences in circadian rhythm or phenology, either in regards to host search or mating behaviors, where close relatives are able to reduce competition by varying their activity cycles (*Mitchell et al., 2015*). Conversely, it has been well-documented that both *D. pseudoobscura* and *D. subobscura* share similar crepuscular activity (*Atkinson and Miller, 1980*; *Bächli et al., 2006*; *Michell and Epling, 1951*; *Noor, 1998*). Thus, at this juncture we do not feel that the circadian rhythms or seasonal activities of feeding or courtship play any distinct role in the observed evolutionary divergence in relative eye or antenna size. As such, while the light-dependent courtship of *D. subobscura* suggests a difference from that of *D. pseudoobscura* in daily patterns of mating, this has not been shown (*Atkinson and Miller, 1980*; *Michell and Epling, 1951*; *Noor, 1998*). Thus, we suggest that it is more likely that *D. subobscura* simply uses consistent visual stimuli as a species-defining trait, perhaps initiated via a preference for a better-lit arena to perform their courtship ritual and to attract a potential mate. This would include such microhabitats as a forest edge or an open forest canopy (*Figure 4C*), where visual elements of courtship such as wing displays would be more optimally employed for species identification and female sexual selection given the increases in light availability. Here, we suggest while these species are assumed to be linked via a forest ecology (*Bächli et al., 2006*; *Burla et al., 1986*; *Michell and Epling, 1951*), that *D. pseudoobscura* may be more likely to prefer darker, inner-forest habitats, while *D. subobscura* would prefer the forest edge or sections of open canopies within the same forest environment (*Figure 4C*). This light preference would therefore create opposing spatial regions of highest abundance, where each species would reduce overlap with the other by tuning their sensory systems towards either larger-eyes and positive phototaxis, or smaller eyes and negative phototaxis. We thus propose that this shift in the nervous system would then affect both courtship and host preference. Here, field studies in the 1980s using baited traps did not find a distinct difference in capture across light versus dark areas for *D. subobscura* (*Atkinson and Miller, 1980*). However, as mentioned before, additional fieldwork is still needed to continue to test our hypotheses outside of the laboratory, and within naturally occurring populations, for example, in relation to abiotic measures of light gradients and GPS studies of forest canopy cover, and in locations where several species co-occur. It is important to note that, to our knowledge, of the 1200–1500 species of *Drosophila* that have been documented, none have been described to be nocturnal (which could be an alternative factor for the

evolution of especially large and light-sensitive eyes). Thus, it is reasonable that increases in eye size for this genus correlate so strongly with positive phototaxis and correlate with visually mediated courtship (*Figure 5F,G*), and we propose this occurrence may extend toward all members of the *Drosophila* genus (*Keesey et al., 2019*). Moreover, the importance of potential visual displays in courtship and predator avoidance has also been previously examined in some Lepidopterans, including phylogenetic studies across spatial and ecological gradients (*Montgomery and Merrill, 2017*), as well as in regards to the genetics of wing pigmentation, especially wing spots (*Zhang and Reed, 2016*). As such, there may be additional factors to address in this interplay between visual and olfactory investment, especially if these same or related genes can be shown to have additional effects beyond the head, for example on the wing pigmentation or across other morphological fodder for evolutionary pressures to exert meaningful sensory changes.

The utilization of forest openings are well studied in avian biology, where males often construct and clear elaborate arenas to perform intricate visual displays for females (i.e. the genus *Parotia* or six-plumed birds of paradise) (*Ligon et al., 2018*). However, to our knowledge, the visual capabilities across vertebrate animal species has never been compared to examine evolutionary investments in the nervous system that correlate with visual courtship, and never in regards to alternative sensory methods of courtship such as olfactory or pheromone driven mating rituals. Again, we feel it is likely that investment in the visual system might mirror the tendency of any *Drosophila* species to possess a positive phototaxis, as all documented species are diurnal. Here we demonstrate that tendency among these five *obscura* species, although it remains unclear which of these behavioral phenotypes occurs first (e.g. phototaxis or visual courtship), and which behavior subsequently drives a correlation in the other trait over the course of evolutionary time. Using our sympatric species (e.g. *D. pseudoobscura* and *D. persimilis*; *Figure 5D,F,G*), we observe that small changes in morphological investment (EF ratio) correlate with dynamic differences in behavior. Here we note that changes in phototaxis between sympatric species appear to be stronger and more acute than changes in courtship dynamics (*Figure 5D,F,G*). For example, we observe that the slope of the correlation between phototaxis and EF Ratio for the sympatric species is greater than the slope related to the rest of the *obscura* subgroup (*Figure 5F*; orange vs grey). Thus, we hypothesize minor shifts in morphology and neurobiology between sensory components of the nervous system first create exaggerated changes in phototaxis behavior, perhaps due to a spontaneous developmental mutation. This might initially separate species spatially and then subsequently, courtship characteristics start to drift apart (e.g. chemical, auditory or visual cues), which ultimately leads to a division that no longer allows successful mating or progeny to occur and thus that incipient species diverge more permanently.

An alternative hypothesis would be that the phototaxis behaviors may have switched before morphology for some species, through an as of yet unknown mechanism, in order to push these species towards more shaded environments (i.e. perhaps to avoid desiccation pressures). As the visual system is inherently expensive to maintain (*Niven and Laughlin, 2008*), this could produce a reduction in the visual system, with the evolutionary pressure being energy preservation. Thus, reduction in visual investment would be a consequence, not a driving force, of niche partitioning. This is similar to what occurred within cavefish, where these species living in complete darkness have lost their eyes entirely. However, it is not thought that the fish went to the caves and then speciated as a consequence of losing their vision, rather that the cavefish specialized only after entering into the cave environment, where the no-longer useful eyes were eventually lost. This concept of the evolutionary order of events goes in a direction that remains puzzling across the animal kingdom. Nocturnal animals usually have two pathways for visual investment, either to increase or decrease the eyes. Here, the selection of the preferred visual investment is also not consistent across nocturnal examples, such as owls (large eyes) compared to bats (small eyes). Thus, the evolutionary decision about visual investment seems to rely on the respective starting point for the species or organism, for example if there already has been large visual or auditory investment, then this is perhaps more prioritized over generations. There may also be additional developmental constraints. Moreover, while these cavefish have entirely lost their eyes, as they are in absolute darkness and therefore represent a very extreme habitat example, many deep-sea fish, in direct contrast, have actually increased their eye size investment (e.g. despite the complete darkness), as they perhaps have to observe all manner of bioluminescence. Thus, overall, this is a difficult evolutionary concept to clarify fully, and in absolute terms, especially without more data from additional animal species. Again, we also point out that zero *Drosophila* species are described as nocturnal, thus variation in eye size may be under

significantly different pressures than nocturnal insects, such as crickets (i.e. Orthopterans), which have often greatly increased their visual investment for their nocturnal activity. Here in the present study, we observe that the more significant changes in phototaxis still occur between close, genetic relatives, which appear prior to significant changes in courtship (*Figure 5D,F,G*). As such, we continue to predict that niche partitioning, character displacement, and response to light gradients are the stronger initial driving forces of the evolution and speciation within the *obscura* group, where the novel environmental conditions drive sensory investment that in turn optimizes the courtship success of each species in this new niche.

It continues to be unclear which are the most important factors in the visual displays of *D. subobscura* during courtship, for example, whether outstretched wings provide a specific color or UV pattern (*Shevtsova et al., 2011*), or whether this wing display simply generates a flash of bright light reflected toward the female (*Figure 3B,D,E*). Moreover, it has been shown that male *D. subobscura* do not sing, and thus do not vibrate their wings during display, but we do observe midleg tapping or drumming, which may instead be the auditory component of their courtship ethology. Further work is still needed to qualify and quantify the courtship variation between these species, especially as it pertains to multimodal sensory integration. Thus far, no research has simultaneously compared visual and auditory neurobiology or development for these species, but future work should attempt to encompass these and other sensory modalities. Additional studies will also need to address which photoreceptors are expanded in the compound eye of *D. subobscura* and how they validate the increases in ommatidium numbers when compared to other close relatives. However, previous research has already shown an expansion of the *fruitless* positive labeled cells in the optic lobes of *D. subobscura* as compared to *D. melanogaster* (*Tanaka et al., 2017*). Thus, while this pathway has not been addressed yet in *D. pseudoobscura* or any other members of the *obscura* group, it is perhaps again indicative of an evolutionary investment in visual modalities for courtship success, given the visual connection to this *fruitless* labeled neural pathway. Moreover, additional studies should address any sensory investment differences between the sexes, especially given that the *fruitless* neural network is sexually dimorphic (*Gaspar et al., 2020*; *Tanaka et al., 2017*).

Nevertheless, it is apparent from our current data that variation in visual and olfactory sensory system development occurs for more than just mating purposes, and appears to match ecological deviations in behavioral phototaxis and microhabitat preferences for light within a shared ecological niche. In the future, it will continue to be important to test our theories related to niche partitioning as an evolutionary force for speciation across other groups beyond *obscura* and to continue to provide ecological explanations for the observed variation or tradeoff between these two sensory systems in relation to other geographical overlap and between competing species across the entire genus. In general, additional work is still needed to qualify and quantify the diverse sensory-driven behaviors across this genus of insects, especially as related to their natural ecology and not just laboratory assays, in order to pave the way for future analyses using genetic resources to identify the neural mechanisms governing these morphological and behavioral variations between sensory systems.

# Materials and methods

## Key resources table

| Reagent type (*species*) or resource | Designation | Source or reference | Identifiers | Additional information |
|---|---|---|---|---|
| Strain (*Drosophila subobscura*) | Dsub 1 | NDSSC | RRID:FlyBase_FBst0203739 | 14011–0131.16 |
| Strain (*D. subobscura*) | Dsub 2 | NDSSC | RRID:FlyBase_FBst0201472 | 14011–0131.04 |
| Strain (*D. subobscura*) | Dsub 3 | NDSSC | RRID:FlyBase_FBst0201473 | 14011–0131.05 |
| Strain (*D. pseudoobscura*) | Dpse 1 | NDSSC | RRID:FlyBase_FBst0200037 | 14011–0121.00 |

*Continued on next page*

*Continued*

| Reagent type (*species*) or resource | Designation | Source or reference | Identifiers | Additional information |
|---|---|---|---|---|
| Strain (*D. pseudoobscura*) | Dpse 2 | NDSSC | RRID:FlyBase_FBst0200038 | 14011–0121.03 |
| Strain (*D. pseudoobscura*) | Dpse 3 | NDSSC | RRID:FlyBase_FBst0201452 | 14011–0121.100 |
| Strain (*D. affinis*) | Daff 1 | NDSSC | RRID:FlyBase_FBst0200081 | 14012–0141.00 |
| Strain (*D. affinis*) | Daff 2 | NDSSC | RRID:FlyBase_FBst0201485 | 14012–0141.05 |
| Strain (*D. affinis*) | Daff 3 | NDSSC | RRID:FlyBase_FBst0203594 | 14012–0141.09 |
| Strain (*D. bifasciata*) | Dbif 1 | KYORIN-fly | | E-12733 |
| Strain (*D. bifasciata*) | Dbif 2 | KYORIN-fly | | E-12701 |
| Strain (*D. bifasciata*) | Dbif 3 | KYORIN-fly | | E-12710 |
| Strain (*D. persimilis*) | Dper 1 | NDSSC | RRID:FlyBase_FBst0200020 | 14011–0111.00 |
| Strain (*D. persimilis*) | Dper 2 | NDSSC | RRID:FlyBase_FBst0200034 | 14011–0111.41 |
| Strain (*D. persimilis*) | Dper 3 | NDSSC | | 14011–0111.63 |
| Other | data repository | EDMOND | https://dx.doi.org/10.17617/3.3v | |

## External morphometrics from head and body

For each fly species, 8–10 females were photographed using a Zeiss AXIO Zoom.V16 microscope (ZEISS, Germany, Oberkochen), including lateral, dorsal, and frontal views. We utilized the following laboratory strains: *D. subobscura* (#1, 14011–0131.16; #2, 14011–0131.04; #3, 14011–0131.05), *D. pseudoobscura* (#1, 14011–0121.00; #2, 14011–0121.03; #3, 14011–0121.100), *D. affinis* (#1, 14012–0141.00; #2, 14012–0141.05; #3, 14012–0141.09), *D. persimilis* (#1, 14011–0111.00; #2, 14011–0111.41; #3, 14011–0111.63), and *D. bifasciata* (#1, E-12733; #2, E-12701; #3, E-12710). Insects were obtained from the National *Drosophila* Species Stock Center (NDSSC, Cornell, USA) or from KYORIN-Fly, the *Drosophila* species stock center at Kyorin University (KYORIN-fly, Tokyo, Japan). We reared all insects with softened standard diet and a single crushed blueberry (in order to further induce and improve egg-laying behaviors). Flies of each wild type were dispatched using pure ethyl acetate (MERCK, Germany, Darmstadt). Lateral body (40×), dissected frontal head (128×), and dissected antenna views (180×) were acquired as focal stacks with a 0.5x PlanApo Z objective (ZEISS, Germany, Oberkochen). The resulting stacks were compiled to extended focus images in Helicon Focus 6 (Helicon Soft, Dominica) using the pyramid method. Based on the extended focus images, we measured head, thoracic, abdominal, foreleg (femur), as well as funiculus and compound eye surface areas, where all measurements are in µm or µm$^2$ (*Figure 1*; *Figure 1—figure supplement 1*). We also measured surface areas of the maxillary palps and length of the ocelli from both species; however, we did not find any significant difference for the palps (*Figure 1C,G*; *Figure 1—figure supplement 2*). Measurements of all body regions were conducted manually using the tools available in Image J (Fiji) software. All raw data available with online version of the manuscript.

## Ommatidia counts and compound eye surface area metrics

In order to count ommatidia, the compound eye of each species was arranged laterally and perpendicular to the AXIO Zoom.V16 microscope. A total of 8–12 individuals per species were utilized, with only the best eight specimens used where the eye was completely intact and in focus, where counts were done manually using Image J (Fiji) software tools (*Figure 1C–E*). We also examined the association between eye surface area and ommatidia counts (*Figure 1D*). Here, we note that species share

nearly identical linear regression analyses between the number of ommatidia and the associated surface area, thus we conclude that ommatidium diameter is identical between the two species, and that surface area is a good predictor of ommatidia number. Although we observed small variations in absolute body size within our species populations that appeared to be correlated with rearing density (e.g. high density produced smaller flies), we also observed a consistently conserved ratio between the eye and antenna morphology regardless of adult body size (data not shown). However, to control for density-dependent plasticity, we maintained both species at a consistent population size (15 females per rearing vial). This resulted in all flies for each species being nearly identical in adult body size for use in morphometric analyses as well as all behavioral examinations. We used the following populations for these measurements: *D. pseudoobscura* #3, *D. persimilis* #1, *D. affinis* #1, D. bifasciata #1, *D. subobscura* #1.

### 3D reconstructions and neuropil measurements

In order to assess neuroanatomy, the dissection of fly brains was carried out according to established protocols (*Keesey et al., 2019*). The confocal scans were obtained using confocal laser scanning microscopy (Zeiss confocal laser scanning microscope [cLSM] 880; ZEISS) using a 40x water immersion objective (W Plan-Apochromat 40×/1.0 DIC M27; ZEISS) in combination with the internal Helium-Neon 543 (ZEISS) laser line. Reconstruction of whole OLs and ALs was done using the segmentation software AMIRA version 5.5.0 (FEI Visualization Sciences Group). We analyzed scans of at least three specimens for each and then reconstructed the neuropils using the segmentation software AMIRA 5.5.0 (FEI Visualization Sciences Group). Using information on the voxel size from the cLSM scans as well as the number of voxels labeled for each neuropil in AMIRA, we calculated the volume of the whole AL as well as the individual sections of the OL and the central brain (where central brain values exclude the AL volume). We used these strains for all measurements: *D. pseudoobscura* #3, *D. subobscura* #1.

### Analyses of courtship and mating behavior

For the analysis of courtship behavior, the adult flies were collected from pupae that were separated into single vials (using a wet paint brush), and then later identified by sex after subsequent eclosion. Adults were kept virgin in these single vials for 2–6 days after eclosion with access to food and water. Temperature controlled chambers were used for courtship conditions. Here we optimized the temperature for both *obscura* species, where courtship initiation and success was observed to be highest between 18–24 degrees Celsius, which was a substantially lower temperature than previous examinations of *D. melanogaster* courtship. In the behavioral assays, a female fly was first aspirated into the tiny chamber, and secured with a clear cover slide (*Figure 1—figure supplement 2G*). Next, a male fly was introduced into the same chamber, and video recording was initiated. The flies were recorded under white light illumination for 10–15 min. If no initiation of courtship was observed after 10 min, then videos were halted and new flies were introduced as a novel pair. Videos of successful courtship and copulation were analyzed with BORIS (http://www.boris.unito.it/). We used the following strains for these behavioral experiments: *D. pseudoobscura* #3, *D. persimilis* #1, *D. affinis* #1, D. bifasciata #1, *D. subobscura* #1.

### Wing interference patterns and pigmentation

In order to assess visual elements of adult wings from both *obscura* species, individual wings from each species were photographed using an AXIO microscope, as was described previously for external head and body metrics. Both clear as well as dark, opaque backgrounds were used to examine wing interference patterns (WIPs) and any other elements of visual information that the wings represent during courtship display (*Figure 3E*). Here, we noted differences in wing shape, as well as sensillum and hair lengths along the wing margins of these two species. However, we did not observe any obvious differences in WIP, nor did we note any apparent differences in pigmentation, color or other visual structures. Thus, it would appear that the wings of the two species are nearly identical, and that perhaps only the behavioral utilization of the wing differs between these species during male courtship (*Figure 1A–D*).

## Phototaxis behavior and Y-tube two-choice experiments

A glass Y-tube was fixed and positioned at approximately a 15-degree slope (which encouraged upward walking), with one arm covered with an opaque cardboard box that was cut to match the diameter of the glass (*Figure 4A*). This covered area provided a heavily darkened arm of the Y-tube, while the other arm was fully illuminated. Both terminal ends of the Y-tube contained sealed glass containers for insect collection and removal. Adults were introduced into the base of the glass Y-tube using an aspirator, where adults could freely walk out of the aspirator pipette tip once they had calmed, and acclimatized to the setup (this greatly reduced escape responses, and random choices). We positioned a light source that mimics natural sunlight wavelengths at the end of the Y-tube, and all overhead illumination (as well as all other sources of light in the chamber) were eliminated. Adult flies were allowed to walk up the Y-tube where they had to then choose between either a dark or light arm, where the first choice was noted for each individual fly (*Figure 4B*), and time duration was also recorded (*Figure 1—figure supplement 1D*). After every 10 individuals, an additional, clean glass Y-tube was used (to avoid any contamination from cuticular hydrocarbons or frass/feces left behind by previous flies [*Keesey et al., 2016*]), and the Y-tubes were rotated after every fly to eliminate any directional bias that could be caused by imperfections in the glass or Y-tube arms. We also rotated the darkened arm every time we exchanged the Y-tube for a clean one, to eliminate any left-right bias. Each day we cleaned glassware with hot soapy water, then rinsed with cold water, then rinsed with ethanol, and lastly we heated them for several hours at 200°C before use in these behavioral assays. In both species, the males showed a stronger trend of light preference than females; however, this trend was not significant (*Figure 4B*). We also noted no significant differences in the time it took flies to make a choice (*Figure 1—figure supplement 1D*), but there was a trend that *D. pseudoobscura* were slighty faster, as were the males of both species when compared to females. We used the following strains for these behavioral experiments: *D. pseudoobscura* #3, *D. persimilis* #1, *D. affinis* #1, D. bifasciata #1, *D. subobscura* #1.

## Statistical assessments and figure generation

All images and drawings are originals, and were prepared by the first author for this publication. Figures were prepared via a combination of R Studio, Microsoft Excel, IrfanView v4.52, ScreenToGif, VLC Media Player, and Adobe Illustrator CS5. Statistics were performed using GraphPad InStat version 3.10 at $\alpha = 0.05$ (*), $\alpha = 0.01$ (**), and $\alpha = 0.001$ (***) levels. Error bars for bar graphs are standard deviation. Boxplots represent the median (bold black line), quartiles (boxes), as well as 1.5 times the inter quartile range (whiskers).

## Supplementary information

All data supporting the findings of this study, including methodology, display examples, raw confocal images and z-stack scans, statistical assessments, courtship videos, as well as other supplementary materials are all available with the online version of this publication. An additional, online data depository also contains raw data from this publication, and this material can be accessed via EDMOND, the Open Access Data Repository of the Max Planck Society (MPG): https://dx.doi.org/10.17617/3.3v.

## Acknowledgements

This research was maintained through funding provided by the German government and the Max Planck Society (Max Planck Gesellschaft). Wild-type flies were obtained from the San Diego *Drosophila* Species Stock Center (now The National *Drosophila* Species Stock Center, Cornell University) as well as obtained from Ehime University at Matsuyama (EHIME-Fly, Japan), which is the branch laboratory for *Drosophila* resources under the National BioResource Project (now moved to Kyorin University, Japan; KYORIN-Fly). We express our gratitude to S Trautheim and D Veit for their technical support, expertise and guidance at MPI-CE. We also thank R Stieber for her expertise and guidance in regards to immuno staining for these two novel fly species. A special thank you as well to J Balma for her expert assistance and proficiency in compiling the curated courtship video examples. Lastly, we thank Ben Longdon, Nicolas Gompel, Benjamin Prud'homme and their laboratories for their gracious donation of several *obscura* species stocks for additional measurements.

## Additional information

### Funding

| Funder | Grant reference number | Author |
| --- | --- | --- |
| Max-Planck-Gesellschaft | Open-access funding | Ian W Keesey<br>Veit Grabe<br>Markus Knaden<br>Bill S Hansson |

The funding organization had no role in the study design, data collection, interpretation, nor the decision to submit the work for publication. The authors declare no competing interests.

### Author contributions

Ian W Keesey, Conceptualization, Resources, Data curation, Formal analysis, Investigation, Visualization, Methodology, Writing - original draft, Project administration, Writing - review and editing; Veit Grabe, Conceptualization, Data curation, Software, Formal analysis, Investigation, Visualization, Methodology; Markus Knaden, Supervision, Validation, Project administration, Writing - review and editing; Bill S Hansson, Supervision, Funding acquisition, Validation, Project administration, Writing - review and editing

### Author ORCIDs

Ian W Keesey (iD) https://orcid.org/0000-0002-3339-7249
Veit Grabe (iD) http://orcid.org/0000-0002-0736-2771
Markus Knaden (iD) https://orcid.org/0000-0002-6710-1071
Bill S Hansson (iD) https://orcid.org/0000-0002-4811-1223

### Decision letter and Author response
Decision letter https://doi.org/10.7554/eLife.57008.sa1
Author response https://doi.org/10.7554/eLife.57008.sa2

## Additional files

### Supplementary files
• Transparent reporting form

### Data availability

All data supporting the findings of this study, including methodology, display examples, raw confocal images and z-stack scans, statistical assessments, courtship videos, as well as other supplementary materials are all available with the online version of this publication. An additional, online data depository also contains raw data from this publication, and this material can be accessed via EDMOND, the Open Access Data Repository of the Max Planck Society (MPG):https://doi.org/10.17617/3.3v.

The following dataset was generated:

| Author(s) | Year | Dataset title | Dataset URL | Database and Identifier |
| --- | --- | --- | --- | --- |
| Keesey IW, Grabe V, Knaden M, Hansson BS | 2020 | Divergent sensory investment mirrors potential speciation via niche partitioning across *Drosophila* | https://doi.org/10.17617/3.3v | EDMOND, 10.17617/3.3v |

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
