## [Decision Letter]

**Acceptance summary:**

This paper elegantly combines extensive morphological and behavioral analyses of five related *Drosophila* species of the obscura group and shows that each species displays a particular set of traits, which can be interpreted all together as differential investments between visual and olfactory sensory systems. Such variation in visual and olfactory investment may have provided relaxed competition and thus facilitated speciation.

**Decision letter after peer review:**

[Editors’ note: the authors submitted for reconsideration following the decision after peer review. What follows is the decision letter after the first round of review.]

Thank you for submitting your work entitled "Niche partitioning as a selective pressure for the evolution of the *Drosophila* nervous system" for consideration by *eLife*. Your article has been reviewed by three peer reviewers, including Virginie Courtier-Orgogozo as the Reviewing Editor and Reviewer #1, and the evaluation has been overseen by a Senior Editor.

Our decision has been reached after consultation between the reviewers. Based on these discussions and the individual reviews below, we regret to inform you that your work cannot be considered at present for publication in *eLife*. However, a completely revised version of this work that take into account the reviewer concerns might be submitted as a new manuscript.

The ideas and concepts presented in this manuscript are very interesting, very well written and perfectly suited to the broad readership of *eLife*. However, the manuscript in its current form is relatively data sparse, in the sense that the ideas are not sufficiently backed up by data.

After consultation, the reviewers have identified 5 major concerns that need to be solved to make the paper suitable for publication in *eLife*:

1) Only one strain was used for each species.

It is important to repeat the measurements for at least one more strain per species, not the ones of the neuropile, which might be too much work, but at least the external ones for eye and antennae size.

2) The treatment of body size should be revised.

3) Ommatidia diameter should be measured, and not just inferred from the measures of ommatidia number and eye area.

4) EF in Figure 5: it is not clear whether EF is significantly different between *D. pseudoobscura* and *D. persimilis*. This should be tested statistically.

5) The title makes a claim that is too strong.

We suggest something like: "Speciation by niche partitioning can be explained by differences in investment between olfactory and visual systems"

Reviewer #1:

This paper examines multiple morphological and behavioral traits in five related species of the obscura group. The authors find that each species display a particular set of traits, which can be interpreted all together as differential investments between visual olfactory sensory systems. These results, obtained in the laboratory using laboratory strains, provide important insights into possible mechanisms of speciation. Here the authors propose based on their data that these species have first diverged in terms of light preference (relative to olfactory investment), and that behavioral separation according to light has led to niche partitioning and courtship deviation.

The figures, text and annotated videos are very clear and nice.

Note that the reviewers have not yet had access the raw data.

Major comments:

1) Only one strain was examined for each species and the authors assume that what they measure in one strain is representative of the entire species. This is problematic because the phenotypic traits measured by the authors can vary between strains within the same species.

The authors should at least acknowledge this caveat in the results and Discussion section.

The authors should also provide a list of the exact strains they used, with information on how long ago they were collected, so that the reader can have information on how long these strains have been maintained in the lab before the experiments were performed.

2) It is not clear at many instances in the manuscript whether measurements are given for males or females or both.

The text mentions that "Eight to ten males and females of these two main species were…" but then in the following sentences it is not clear if the results are presented for males, females or both.

The authors should clarify this point for Figure 1A,B,C,D,E,F,G Figure 2A,B,C,D,E Figure 5A,B,C,D,E

3) "In these sympatric species we see larger behavioral variations (y-axis) than changes in morphology (x-axis)" and "we note that changes in phototaxis appear to be stronger and more acute than changes in courtship dynamics between sympatric species".

How can you compare traits which are measured in different units? This is unclear.

Reviewer #2:

Keesey et al. put forward an interesting hypothesis. That closely related *Drosophila* species that are sympatric might have diverged by partitioning their niche, with one species preferring light conditions similar to those found at the edge of the forest and at clearings, while the other species would prefer darker environments. They argue that these behavioural differences are driven by different investment in the visual vs the olfactory systems. The main novelty of the paper is to quantify the size of the eye and antennae, as well as their associated neuropils across four closely related *Drosophila* species, and correlate these differences with two behaviours: courtship and phototaxis. Based on this, they hypothesise that differential allocation of resources to vision and olfaction would underlie these differences in behaviour, and thus drive niche partitioning. However, their data is not sufficient to support this claim. Given the data presented, the title of the manuscript seems misleading: "Niche partitioning as a selective pressure for the evolution of the *Drosophila* nervous system" , suggests that they provide some experimental evidence to demonstrate that niche partitioning acts as a selective pressure for nervous system evolution. However, they do not demonstrate at any point that there is niche partitioning in the wild for these species, or that competition from sympatric species imposes a selective pressure on the evolution of the eyes and antennae. Therefore, a title that would be more apt for what the manuscript shows would be what they suggest as impact statement: "Phototaxis and courtship behaviour match differences in olfactory and visual system investment in five monophyletic *Drosophila* species and could explain their speciation events". In addition, there are a number of statements through the text that are overinterpretations of the presented results. I will mention some these in the below in addition to other concerns:

– In Figure 1, why are the authors not showing the measurements normalised by body size? Or head capsule size? Especially since in Figure 2 they normalise by adult size. Why do they use total size normalised measurements in some figures, but not in others?

– In Figure 5 D and E the correlation R values are shown without p values. There is also no statics shown for the data in Figure 5B.

– Their explanation for the data shown in Figure 5 was unconvincing. The fact that *D. pseudoobscura* and *D. persimilis* have very similar EF ratios (7.7 vs 7.8, is this difference even significant?) yet display different phototaxis and courtship behaviour, could reflect that their hypothesis is wrong, and that different resource allocation between vision and olfaction does not underlay behavioural differences, and might just be a by-product of the true (unknown) mechanistic changes underlying this behavioural evolution. Instead, they try to argue that this data shows that "small changes in morphological investment (EF ratio) can create dynamic differences in behaviour". However, there is no evidence to support this statement.

– In addition, their claim that the niche partition, and the differential allocation to vision and smell, occurs as a consequence of competition, is based on data from *D. pseudoobscura* and *D. subobscura*. However, these two species are not sympatric, the first one inhabits in North America, while the second one in Europe. At the same time, the two species that are sympatric, *D. pseudoobscura* and *D. persimilis*, have different behaviours, yet, very similar (identical?) allocation to vision and olfaction. How does this fit with their hypothesis?

– Without any fieldwork to show that there is indeed niche partitioning in the light and dark areas of the forest for the two main species studied, any claims of this nature need to be removed from the manuscript, specially the title and the Abstract.

Reviewer #3:

This paper explores external eye and antennal morphology as well as the associated internal nervous system morphology involved in the function of these important sensory organs between *D. subobscura* and *D. pseudoobscura* as well as several related species. They report that *D. subobscura* has larger eyes and smaller antennae than *D. pseudoobscura*. Comparisons of the behavior of these flies suggest that these changes in the relative size of sensory structures are correlated with differences in copulatory behavior and phototatic preferences. These correlations are also consistent with the morphology and behavior of related species.

Overall the manuscript provides a very interesting study of the potential behavioral outcomes of changes in the size of eyes and antennae, and that this may contribute to habitat preferences and even mating differences. However, I have a number of concerns about the manuscript that should be addressed.

Substantive concerns

In addition to ommatidia number, ommatidia size/diameter is a key parameter for the function of compound eyes. The paper states that "ommatidial diameter was identical" between the two focal species. However as far as I can tell this was never directly measured but instead was inferred from the correlation between ommatidia number and eye area. This is problematic for several reasons – ommatidia diameter can vary considerably across individual eyes and therefore even if there was a perfect correlation between number and area (which there is not) this could still belie local but important differences in ommatidia diameter for example in the anterior ommatidia. In addition, while Figure 2D shows a similar positive correlation between ommatidia number and area within *D. subobscura* or *D. pseudoobscura*, the species actually differ in that the former has smaller than expected eye area compared to ommatidia number and vice versa for the latter species. It is likely that this could be explained by *D. pseudoobscura* having larger ommatidia. This is important because it would potentially confer greater contrast sensitivity and explain why *D. pseudoobscura* may prefer darker conditions.

Although the manuscript states "Eight to ten males and females of these two main species…", no data is presented for males. Given there is interesting patterns of eye and antennal dimorphism between sexes for many Drosophla species, and this manuscript studies copulatory behavior and phototatic preferences, I think it is important to include data from males and as well as females, but no explanation is provided for why only female morphology was analysed.

The authors make much of the correlation between external and internal morphology and copulatory and phototatic behaviors of the two focal species and related species. However, I am concerned that only one strain of each species was examined and given the substantial intra-specific variation, and even plasticity in these traits, it might be misleading to say that these differences and correlations are indicative of these species (with the ensuing speculation about speciation etc) when it might only be representative of the strains used.

The paper does not cite other previous studies showing a trade-off between eyes and other aspects of the head capsule in *D. pseudoobscura* and in other *Drosophila* species and especially where the genetic underpinnings have been explored in some detail – in particular the recent paper in Dev Cell by Ramaekers et al., as well as work from the Norry and McGregor groups.

[Editors’ note: further revisions were suggested prior to acceptance, as described below.]

Thank you for submitting your article "Divergent sensory investment mirrors potential speciation via niche partitioning across *Drosophila*" for consideration by *eLife*. Your article has been reviewed by three peer reviewers, including Virginie Courtier-Orgogozo as the Reviewing Editor and Reviewer #1, and the evaluation has been overseen by Ronald Calabrese as the Senior Editor.

The reviewers have discussed the reviews with one another and the Reviewing Editor has drafted this decision to help you prepare a revised submission.

This manuscript reports very interesting data on trade-offs between olfactory and visual organs and behaviours in flies of the obscura group of *Drosophila*. The authors have extensively revised the manuscript in response to the reviewers' comments on the previous version of the manuscript. The new additional data (e.g. in Figures 5 and Figure 1—figure supplement 3) very much enhance the manuscript and add further support to the authors conclusions. While, reviewer 3 thinks that it would be interesting to study ommatidial size in different regions of the eyes of these flies and to assay males as well as females, reviewer 3 agrees with the authors that this can be done in the future and is not needed for this story because the conclusions are already very well supported by the data provided.

1) One caveat highlighted by the reviewers was that the phenotypic traits measured by the authors can vary between strains within the same species. As far as I understand, in the revised manuscript, the surface areas of the compound eye, antenna, maxillary palps, ocelli, and overall body size, as well as head, thorax, abdomen and femur length are still presented for only one strain per species.

However, the authors replied to the reviewers' comments that they included data from 3 populations/strains for each of our 5 total obscura species. And the revised Discussion reads: "the present study, which documents this sensory inversion across the obscura group, including across close, sympatric relatives, as well as across several populations or strains of each species (Figure 5A,B)."

Can they clarify this point?

If data is available, it would be good to increase the number of strains for Figure 1. If not, it is recommended to acknowledge in the text that it remains to be shown whether the observed differences are truly interspecific, as opposed to intraspecific, by analyzing several strains for each species.

2) Figure 5:

Where do the data in Figure 5B come from? From the x axis labels it seems that the authors choose a single strain for the comparison, but the data do not match the data shown in A. See for example the data distribution for Dpse1 or Dsub1 in panel A and B, they are completely different. Also it might be best to plot in B the aggregated measures of all three strains, unless, is this what they did? If that is the case they need to correct the axis labels, which at the moment indicate that only strain 1 for each species was used.

Assuming that the problem with Figure 5B are the labels and that indeed this represents the aggregated data for each of the three strains per species. This looks more convincing and is fine. However, it is worth noting that intraspecies variability seems larger than the interspecies variability for *D.pseudoobscura* and *D. persimilis*. This could give them an opportunity to test their hypothesis further. For example, unlike the population average, Dpse2 (EF ratio 8.16) is higher than Dper (EF ratio 7.91), if they examine the behaviour of these two strains, do they find that it correlates as well, ie. in this case Dper prefers darkness and does less frontal courtship than Dpse (unlike what happens with other strains)? This experiment could be done, or at least commented in the Discussion.

3) In addition to this, there seems to be a few extra mistakes in Figure 5 with the numbers. First, in the original publication the EF ratio of Dpse was 7.74 and that of Dper 7.89. However, in the new submission, for none of the measures for any of the strains there is a match on these values. This is surprising as one would have expected one of the strains for each species to match the numbers of Figure 5B from the prior submission, can the authors explain why this is not the case? Did they re-count the EF ratio for all of the strains including the one they had used previously and the different numbers reflect just variability? Can they confirm which one was the strain they had used previously?

4) In the current Figure 5B the ratio for Dper reads 8.20, but in Figure 5D for the same species reads 8.28. I imagine this is just a typo, can they correct this prior to publication?

5) Discussion:

The authors should properly describe existing references in the Discussion. For example, the paper from Atkinson and Miller 1980 shows that in field experiment capture bait experiments, *D. subobscura* did not have a preference for baits located in a light vs dark areas. Meaning that field experiments so far do not support their hypothesis, this should be noted in the Discussion.

6) Also in the Discussion, the statement should change from "changes in morphological investment can create dynamic differences in behaviour" to "changes in morphological investment correlate with dynamic differences in behaviour".

7) In the very extensive Discussion, they don't mention the alternative more plausible, hypothesis that behaviour might have switched first, through unknown neural basis, for some species to prefer shaded places (perhaps to avoid desiccation?) and thus become less visually guided. As the visual system is expensive to maintain this would produce a reduction in the visual system, the evolutionary pressure being energy preservation. Thus, reduction in visual investment being a consequence, not a driving force, of niche partitioning. This is indeed what happens with cave fish, where cave species have lost their eyes all together, but it is not thought that the fish went to the caves and speciated as a consequence of losing their eyes (as the authors seem to be suggesting in their discussion for flies), rather they specialised in the caves, and the no-longer useful eyes were lost. This seems a more parsimonious explanation for differential investment in vision and olfaction, than the proposed mechanism where reduced vision investment would drive niche partitioning. Not to mention that there is no evidence, nor circuit basis that could explain how larger eyes change phototaxis preference, which is probably computed in downstream circuits. Therefore, they should include in their Discussion the very possible hypothesis that flies reduced their visual investment as a consequence of niche partitioning, which occurred through yet unknown mechanisms.

[Editors' note: further revisions were suggested prior to acceptance, as described below.]

Thank you for resubmitting your article "Divergent sensory investment mirrors potential speciation via niche partitioning across *Drosophila*" for consideration by *eLife*. Your revised article has been reviewed by the Reviewing Editor and Ronald Calabrese as the Senior Editor.

The reviewers' comments have been adequately addressed. Figure 5 is now perfect. The Discussion has been greatly improved and is indeed extremely interesting.

There is just one comment that has not been fully addressed: the authors replied that they clarified the point that it remains to be shown whether the observed differences are truly interspecific but I could not read any statement regarding this fact in the Discussion. We suggest to add the following sentence (or similar) to the Discussion: "Please note that surface areas of the various head and thorax organs, as well as overall body size, were examined only for one strain per species. It thus remains to be confirmed that the observed differences are truly interspecific."

Regarding the raw data supporting the findings of this study:

– The raw data for Figure 3C, 4B is missing.

– The standard deviation for AL and OL of *D. pseudoobscura* was not calculated correctly based on the provided Excel file (see cells C45-F45 in Figure 2—source data 1).

– It is really nice that the authors have made the data supporting the findings of this study available through EDMOND, the Open Access Data Repository of the Max Planck Society. However, not all their data is available on this platform. It would be great if the authors could also include all the videos analysed for this paper.

---

## [Author Response]

[Editors’ note: the authors resubmitted a revised version of the paper for consideration. What follows is the authors’ response to the first round of review.]

The ideas and concepts presented in this manuscript are very interesting, very well written and perfectly suited to the broad readership of eLife. However, the manuscript in its current form is relatively data sparse, in the sense that the ideas are not sufficiently backed up by data.After consultation, the reviewers have identified 5 major concerns that need to be solved to make the paper suitable for publication in eLife:1) Only one strain was used for each species.It is important to repeat the measurements for at least one more strain per species, not the ones of the neuropile, which might be too much work, but at least the external ones for eye and antennae size.

This is a very good point, and we have now included data from 3 populations/strains for each of our 5 total *obscura* species (thus for a total of 15 population assessments). This additional data is now available in the new version of Figure 5A,B. In addition, we confirm statistically that the differences in eye-to-funiculus ratio (EF Ratio) are consistent within a species, as well as consistently different between species.

2) The treatment of body size should be revised.

We have now demonstrated again that in the *obscura* group sensory traits also scale isometrically with respect to head size (see Figure 1—figure supplement 1B). Moreover, we have likewise shown this lack of allometry for both head and body sizes across 62 different species in a previous publication, including multiple regression analyses as well as analyses of the residuals (linked DOI below). This is a major justification for using EF Ratio to compare sensory systems between insect species that differ in absolute size. However, we have also made further careful referrals to body size considerations within the present manuscript, including for both the methods and Discussion sections.

(Please see Supplementary Figure 1B,C,E,F,H,I within the following publication)

Keesey I. W., Grabe V., Gruber L., Koerte S., Obiero G. F. et al. , (2019). Inverse resource allocation between vision and olfaction across the genus *Drosophila*. Nature Communications. 10: 1162 10.1038/s41467-019-09087-z

3) Ommatidia diameter should be measured, and not just inferred from the measures of ommatidia number and eye area.

Thank you for this suggestion! We have now added repeated measures of ommatidia diameter from each of the 5 species (e.g. 5 measures per individual replicate, per species, for 25 total diameter measurements per species). This is now available in Figure 1—figure supplement 3. Here we did not identify any difference in diameter between our insects. Again, we would also like to highlight that all raw images, confocal scans, measurements, and other raw data will be made freely available with the online version of this publication for any additional comparisons. Please also see additional comments about ommatidia diameter within response towards Review #3.

4) EF in Figure 5: it is not clear whether EF is significantly different between D. pseudoobscura and D. persimilis. This should be tested statistically.

We have now explicitly tested this difference between species in the new version of Figure 5B, where we confirm that the EF Ratio is significantly different between these two sympatric species.

5) The title makes a claim that is too strong.We suggest something like: "Speciation by niche partitioning can be explained by differences in investment between olfactory and visual systems"

We thank the editors and reviewers for their suggestions, and hope the new title is more indicative of the scope and design of the research presented in this manuscript. The new title is listed as: “Divergent sensory investment mirrors potential speciation via niche partitioning across *Drosophila*”

Reviewer #1:This paper examines multiple morphological and behavioral traits in five related species of the obscura group. The authors finds that each species display a particular set of traits, which can be interpreted all together as differential investments between visual olfactory sensory systems. These results, obtained in the laboratory using laboratory strains, provide important insights into possible mechanisms of speciation. Here the authors propose based on their data that these species have first diverged in terms of light preference (relative to olfactory investment), and that behavioral separation according to light has led to niche partitioning and courtship deviation.The figures, text and annotated videos are very clear and nice.Note that the reviewers have not yet had access the raw data.Major comments:1) Only one strain was examined for each species and the authors assume that what they measure in one strain is representative of the entire species. This is problematic because the phenotypic traits measured by the authors can vary between strains within the same species.The authors should at least acknowledge this caveat in the results and Discussion section.

Thank you for this suggestion! We have now strived these last few months to include 3 populations/strains across each of the 5 *obscura* species that we examine in this manuscript (i.e. for a total of 15 populations examined). This data is now presented in Figure 5A,B. We believe that this additional population data further strengthens the publication, as we now more clearly demonstrate that eye-to-funiculus ratio (EF Ratio), while still variable between strains, is relatively stable within a species, but more divergent between each species. Moreover, we have added a mention of this population-related caveat to the Discussion section of the text.

The authors should also provide a list of the exact strains they used, with information on how long ago they were collected, so that the reader can have information on how long these strains have been maintained in the lab before the experiments were performed.

We apologize for this error. In the new version of the manuscript, we have now sought to include increased levels of detail about the exact strains and populations used for each species. This includes stock numbers that can in turn provide additional information about site of collection, as well as the date since laboratory establishment via the stock centers that have provided all this background. Please see the Materials and methods section.

2) It is not clear at many instances in the manuscript whether measurements are given for males or females or both.The text mentions that "Eight to ten males and females of these two main species were…" but then in the following sentences it is not clear if the results are presented for males, females or both.The authors should clarify this point for Figure 1A,B,C,D,E,F,G Figure 2A,B,C,D,E Figure 5A,B,C,D,E

Thank you for this comment and suggestion. We now clarify the sex represented in each data set and within associated figure legends.

3) "In these sympatric species we see larger behavioral variations (y-axis) than changes in morphology (x-axis)" and "we note that changes in phototaxis appear to be stronger and more acute than changes in courtship dynamics between sympatric species".How can you compare traits which are measured in different units? This is unclear.

We have clarified this section in the text as well as Figure 5F. Here we are comparing the metrics for sympatric species to the overall *obscura* species regression, highlighting the much higher slope for sympatric species.

Reviewer #2:Keesey et al. put forward an interesting hypothesis. That closely related Drosophila species that are sympatric might have diverged by partitioning their niche, with one species preferring light conditions similar to those found at the edge of the forest and at clearings, while the other species would prefer darker environments. They argue that these behavioural differences are driven by different investment in the visual vs the olfactory systems. The main novelty of the paper is to quantify the size of the eye and antennae, as well as their associated neuropils across four closely related Drosophila species, and correlate these differences with two behaviours: courtship and phototaxis. Based on this, they hypothesise that differential allocation of resources to vision and olfaction would underlie these differences in behaviour, and thus drive niche partitioning. However, their data is not sufficient to support this claim. Given the data presented, the title of the manuscript seems misleading: "Niche partitioning as a selective pressure for the evolution of the Drosophila nervous system" , suggests that they provide some experimental evidence to demonstrate that niche partitioning acts as a selective pressure for nervous system evolution. However, they do not demonstrate at any point that there is niche partitioning in the wild for these species, or that competition from sympatric species imposes a selective pressure on the evolution of the eyes and antennae. Therefore, a title that would be more apt for what the manuscript shows would be what they suggest as impact statement: "Phototaxis and courtship behaviour match differences in olfactory and visual system investment in five monophyletic Drosophila species and could explain their speciation events". In addition, there are a number of statements through the text that are overinterpretations of the presented results. I will mention some these in the below in addition to other concerns:

Thank you again for your time and insights with regard to this manuscript. We concur, that in the present document we do not explore field-sampling or other natural monitoring of these *obscura* species. Moreover, with your comments in mind, we have more explicitly added this caveat to the manuscript Discussion section. However, we contend that the theory we put forward and the evidence we provide is very compelling, and thus far, our theory is highly consistent with the data that we have collected in the laboratory using lab-reared flies from our five species. In the future, we hope to continue to test this hypothesis, for example in the field, but this was not within the feasibility nor the scope of the present manuscript revision. In accordance with your comments, and those from the editors, we have adjusted the title and we hope this new version is more agreeable. Thank you again for your time, insights and suggestions!

– In Figure 1, why are the authors not showing the measurements normalised by body size? Or head capsule size? Especially since in Figure 2 they normalise by adult size. Why do they use total size normalised measurements in some figures, but not in others?

We hope we have addressed this concern, as we have now added tests of allometry to Figure 1—figure supplement 1B. Herein we demonstrate that ommatidia number, for example, does not scale with respect to head size; moreover, in a previous publication we examine, through multiple regression and assessments of residuals across 62 *Drosophila* species, that neither body size nor head size are significantly linked to the observed variation in either visual or olfactory sensory investment. In Figure 2 we normalize using the hemisphere (grey) of each species. This is comparable to our normalization via eye-to-funiculus ratio (EF Ratio), which again, we feel best provides statistically comparable values between species of differing absolute size. However, some metrics, like sheer ommatidia counts, do not appear available from the literature, thus we also wanted to include raw values.

(Please see Supplementary Figure 1B,C,E,F,H,I within the following publication):

Keesey I. W., Grabe V., Gruber L., Koerte S., Obiero G. F. et al. , (2019). Inverse resource allocation between vision and olfaction across the genus *Drosophila*. Nature Communications. 10: 1162 10.1038/s41467-019-09087-z

– In Figure 5 D and E the correlation R values are shown without p values. There is also no statics shown for the data in Figure 5B.

Corrected, thank you for catching this error! We have now shown the R^2^ and p-values for each linear correlation, as well as added mention of statistical assessments to the figure legends. Apologies again for this oversight!

– Their explanation for the data shown in Figure 5 was unconvincing. The fact that D. pseudoobscura and D. persimilis have very similar EF ratios (7.7 vs 7.8, is this difference even significant?) yet display different phototaxis and courtship behaviour, could reflect that their hypothesis is wrong, and that different resource allocation between vision and olfaction does not underlay behavioural differences, and might just be a by-product of the true (unknown) mechanistic changes underlying this behavioural evolution. Instead, they try to argue that this data shows that "small changes in morphological investment (EF ratio) can create dynamic differences in behaviour". However, there is no evidence to support this statement.

During the revision of this manuscript, we have now added 2 additional populations from each of the five species (thus 3 examined populations per species, for a total analysis of 15 populations). We feel that this additional, robust data more strongly supports our hypotheses, as well as strengthens the statistical tests between species. Here we contend that EF Ratio is relatively consistent between populations within a species, as well as consistently divergent between our species. Please see the new Figure 5A,B. Moreover, the new population-based EF Ratio for each species now correlates even more strongly with both phototaxis and frontal courtship behaviors (see Figure 5F,G). Again, we thank the reviewers for suggesting the additional population-based analyses of each species, as we feel these additional sets of replicates and statistical analyses have continued to reinforce our theories. While we concur that we cannot eliminate all other alternative hypotheses, nor that we can we pin down causation, we again contest that our hypothesis of a sensory tradeoff is currently the most consistent, viable explanation for the observed behavioral variations in phototaxis and courtship. Here again, we show repeated, strong, statistically correlated evidence within the laboratory, and hope in the future to examine these hypotheses again in the wild, with naturally co-occuring species of flies.

– In addition, their claim that the niche partition, and the differential allocation to vision and smell, occurs as a consequence of competition, is based on data from D. pseudoobscura and D. subobscura. However, these two species are not sympatric, the first one inhabits in North America, while the second one in Europe. At the same time, the two species that are sympatric, D. pseudoobscura and D. persimilis, have different behaviours, yet, very similar (identical?) allocation to vision and olfaction. How does this fit with their hypothesis?

Thank you again for comments and suggestions. We have provided additional clarification in the written text that we did not consider *D.pseudoobscura* and *D.subobscura* to be evolutionarily sympatric (though they are now published as co-occurring in the Western USA, and would make for a great field study as a follow-up!). Moreover, we have provided much stronger data (and population replicates) to support the notion that *D.pseudoobscura* and *D.persimilis* differ in EF Ratio morphology as well as behavior (where these two species are in fact, well recognized as sympatric). While again we concede that is it not possible to pin down causation, we do find the correlation between EF Ratio and phototaxis (as well as the correlation between EF Ratio and frontal courtship) to be even more strongly supported statistically given the increased replicates of measurements per insect species across populations (please see revised Figure 5). Again, we thank the reviewers for their insightful suggestions of adding populations for each *obscura* species, and we hope the new data and text clarifications more convincingly demonstrate our interpretations of the data as compared to the previous manuscript draft.

– Without any fieldwork to show that there is indeed niche partitioning in the light and dark areas of the forest for the two main species studied, any claims of this nature need to be removed from the manuscript, specially the title and the Abstract.

We have attempted to tone down our conclusions about “ecological” niche partitioning (i.e. to concluding that niche partitioning would be consistent with the observed behavioral patterns from the laboratory, and we have added discussion of what further data would be needed to test more conclusively this hypothesis in the field in a future study). While we do not consider this manuscript to be the alpha and omega on this particular research topic, we do feel that the present manuscript provides a strong foundation concerning a viable ecological explanation for inverse variation in the visual and olfactory systems across these closely related insect species, and that this tradeoff has distinct behavioral ramifications. We hope this manuscript could serve as a foothold and building block to continue to examine these hypotheses, perhaps including fieldwork. Further discussion is now added in the text to propose these additional studies, highlighting the importance of field-based validation of our hypotheses towards niche partitioning and species competition within the lighting architecture of a temperate forest environment. Thank you again for your comments, insights and suggestions, where we hope our newly written version and copious new data are more convincing of these hypotheses.

Reviewer #3:This paper explores external eye and antennal morphology as well as the associated internal nervous system morphology involved in the function of these important sensory organs between D. subobscura and D. pseudoobscura as well as several related species. They report that D. subobscura has larger eyes and smaller antennae than D. pseudoobscura. Comparisons of the behavior of these flies suggest that these changes in the relative size of sensory structures are correlated with differences in copulatory behavior and phototatic preferences. These correlations are also consistent with the morphology and behavior of related species.Overall the manuscript provides a very interesting study of the potential behavioral outcomes of changes in the size of eyes and antennae, and that this may contribute to habitat preferences and even mating differences. However, I have a number of concerns about the manuscript that should be addressed.Substantive concernsIn addition to ommatidia number, ommatidia size/diameter is a key parameter for the funntion of compound eyes. The paper states that "ommatidial diameter was identical" between the two focal species. However as far as I can tell this was never directly measured but instead was inferred from the correlation between ommatidia number and eye area. This is problematic for several reasons – ommatidia diameter can vary considerably across individual eyes and therefore even if there was a perfect correlation between number and area (which there is not) this could still belie local but important differences in ommatidia diameter for example in the anterior ommatidia. In addition, while Figure 2D shows a similar positive correlation between ommatidia number and area within D. subobscura or D. pseudoobscura, the species actually differ in that the former has smaller than expected eye area compared to ommatidia number and vice versa for the latter species. It is likely that this could be explained by D. pseudoobscura having larger ommatidia. This is important because it would potentially confer greater contrast sensitivity and explain why D. pseudoobscura may prefer darker conditions.

Thank you for your insights regarding ommatidia. While we measured ommatidia diameter in a previous publication (and show it does not generally vary, despite substantial changes in ommatidia number), we did error in our extrapolation towards all our current *obscura* species. We have now added a new supplementary figure to address direct measurements of ommatidia diameter for each species presented in the current manuscript (please see Figure 1—figure supplement 3). However, here we again do not identify any significant variance in diameter between our tested species, suggesting that surface area and ommatidia number are the more accurate metrics for estimating divergence in visual investment via external morphology. Moreover, I would emphasize again that we are providing all raw data, scans, confocal images and replicates within the accepted version of the present manuscript. We would thus encourage the reviewer and others to examine additional ideas or avenues that we may have overlooked, for example, any additional descriptive measures of ommatidium curvature, apex or length.

We would also add that, to our knowledge, there are no nocturnal *Drosophila* species. Thus while some insect orders and families are more variable in their ommatidia measures, these differences could also be related to circadian periodicity of the animal’s activity patterns. However, all examined *Drosophila* show the same pattern of activity. That all being said, we continue to be highly interested in understanding more about the visual capabilities of this genus of fly, and we would welcome any additional comments or ideas about how to better test, quantify and describe variation in eye size between our species of interest. In the future, we also hope genetic tools become more readily available across the *obscura* clade in order to generate additional markers to follow.

Although the manuscript states "Eight to ten males and females of these two main species…", no data is presented for males. Given there is interesting patterns of eye and antennal dimorphism between sexes for many Drosophla species, and this manuscript studies copulatory behavior and phototatic preferences, I think it is important to include data from males and as well as females, but no explanation is provided for why only female morphology was analysed.

We sought to examine, in detail, many aspects of external and internal investment in visual and olfactory machinery across 5 species and ultimately 3 populations per species (for a new total of 15 insect strains). As such, we regrettably were not able to also assess all differences related to sex. In the present manuscript, we now clarify in the text that we chose to focus on sensory investment for females, which act as the receivers of male courtship displays, and also represent the oviposition decision-makers. However, we would encourage future research to also examine variance related to sex, and we would be excited to see those results in comparison to the current literature!

The authors make much of the correlation between external and internal morphology and copulatory and phototatic behaviors of the two focal species and related species. However, I am concerned that only one strain of each species was examined and given the substantial intra-specific variation, and even plasticity in these traits, it might be misleading to say that these differences and correlations are indicative of these species (with the ensuing speculation about speciation etc) when it might only be representative of the strains used.

Thank you again for your insights. We now examine populations (in triplicates) from each species, as well as include the stock numbers and associated information from each strain. Here we emphasize that the new data more strongly defines the intra- and interspecies differences across these sensory investments, where again, we observe more consistent values within a species than between our species. Please see the new Figure 5 for all new data related to population-based analyses of EF Ratio.

The paper does not cite other previous studies showing a trade-off between eyes and other aspects of the head capsule in D. pseudoobscura and in other Drosophila species and especially where the genetic underpinnings have been explored in some detail – in particular the recent paper in Dev Cell by Ramaekers et al., as well as work from the Norry and McGregor groups.

Apologies for this oversight. We have now included several of the suggested citations, as well as the newest publication showing that this inverse variation (or trade-off) between vision and olfaction occurs simultaneously:

Özer I, Carle T (2020) Back to the light, coevolution between vision and olfaction in the “Dark-flies” (*Drosophilamelanogaster*). PLoS ONE 15(2): e0228939. https://doi.org/10.1371/journal.pone.0228939

Ramaekers, A., A. Claeys, M. Kapun, E. Mouchel-Vielh, D. Potier et al., (2019) Altering the Temporal Regulation of One Transcription Factor Drives Evolutionary Trade-Offs between Head Sensory Organs. Dev Cell 50: 780–792 e787. https://doi.org/10.1016/j.devcel.2019.07.027https:// doi.org/10.1016/j.devcel.2019.07.027

Gaspar, P. et al. (2020) Characterization of the Genetic Architecture Underlying Eye Size Variation Within *Drosophilamelanogaster* and *Drosophila* simulans. G3 (Bethesda, Md.); 10.1534/g3.119.400877

[Editors’ note: what follows is the authors’ response to the second round of review.]

This manuscript reports very interesting data on trade-offs between olfactory and visual organs and behaviours in flies of the obscura group of Drosophila. The authors have extensively revised the manuscript in response to the reviewers' comments on the previous version of the manuscript. The new additional data (e.g. in Figures 5 and Figure 1—figure supplement 3) very much enhance the manuscript and add further support to the authors conclusions. While, reviewer 3 thinks that it would be interesting to study ommatidial size in different regions of the eyes of these flies and to assay males as well as females, reviewer 3 agrees with the authors that this can be done in the future and is not needed for this story because the conclusions are already very well supported by the data provided.1) One caveat highlighted by the reviewers was that the phenotypic traits measured by the authors can vary between strains within the same species. As far as I understand, in the revised manuscript, the surface areas of the compound eye, antenna, maxillary palps, ocelli, and overall body size, as well as head, thorax, abdomen and femur length are still presented for only one strain per species.However, the authors replied to the reviewers' comments that they included data from 3 populations/strains for each of our 5 total obscura species. And the revised Discussion reads: "the present study, which documents this sensory inversion across the obscura group, including across close, sympatric relatives, as well as across several populations or strains of each species (Figure 5A,B)."Can they clarify this point?If data is available, it would be good to increase the number of strains for Figure 1. If not, it is recommended to acknowledge in the text that it remains to be shown whether the observed differences are truly interspecific, as opposed to intraspecific, by analyzing several strains for each species.

We have added complete EF ratio data for all populations to the manuscript in Figure 5A; however, we were not able to generate the complete data comparisons as in Figure 1 for all populations. Moreover, clarification has been added to the Discussion. In addition, we have now been able to provide all data supporting the findings of this study, including methodology examples, raw images and z-stack scans, ommatidia measurements, statistical assessments as well as species and population datasets, which are available through EDMOND, the Open Access Data Repository of the Max Planck Society. We have now completed the necessary steps to make this data open access to the public, and the web link or DOI should be fully functional and accessible when the manuscript is published online.

EDMOND, link for all raw data:

https://dx.doi.org/10.17617/3.3v

2) Figure 5:Where do the data in Figure 5B come from? From the x axis labels it seems that the authors choose a single strain for the comparison, but the data do not match the data shown in A. See for example the data distribution for Dpse1 or Dsub1 in panel A and B, they are completely different. Also it might be best to plot in B the aggregated measures of all three strains, unless, is this what they did? If that is the case they need to correct the axis labels, which at the moment indicate that only strain 1 for each species was used.Assuming that the problem with Figure 5B are the labels and that indeed this represents the aggregated data for each of the three strains per species. This looks more convincing and is fine. However, it is worth noting that intraspecies variability seems larger than the interspecies variability for D.pseudoobscura and D. persimilis. This could give them an opportunity to test their hypothesis further. For example, unlike the population average, Dpse2 (EF ratio 8.16) is higher than Dper (EF ratio 7.91), if they examine the behaviour of these two strains, do they find that it correlates as well, ie. in this case Dper prefers darkness and does less frontal courtship than Dpse (unlike what happens with other strains)? This experiment could be done, or at least commented in the Discussion.

Thank you. Yes. The labels were incorrect in Figure 5B, which, as you already correctly identified, shows the aggregate/average of ALL data across populations within the complete dataset from each species. We have adjusted the figure labels in the Figure 5B accordingly, and thank you again for catching this error! Given the pandemic and laboratory constraints, we are unable to run the additional experiments suggested, but we hope something similar could be constructed in the future to continue to examine these species (and populations) in more detail. We are very grateful for this proposed experiment, and look forward to testing these ideas perhaps in a future manuscript or study! We also add note of this idea to the manuscript.

3) In addition to this, there seems to be a few extra mistakes in Figure 5 with the numbers. First, in the original publication the EF ratio of Dpse was 7.74 and that of Dper 7.89. However, in the new submission, for none of the measures for any of the strains there is a match on these values. This is surprising as one would have expected one of the strains for each species to match the numbers of Figure 5B from the prior submission, can the authors explain why this is not the case? Did they re-count the EF ratio for all of the strains including the one they had used previously and the different numbers reflect just variability? Can they confirm which one was the strain they had used previously?

Apologies. We have added more replicates to several of the species, including those from the original populations. Thus, the average values may have shifted slightly from the previous manuscript version. For each section within the Materials and methods, we document, often as the last sentence, which species and populations we used within each behavioral or morphological analysis. In addition, we have now reduced all numerical averages to only two decimal places, thus we further hope this avoids any confusion related to averaging of the EF values in Figure 5. In general, we hope these comments and adjustments sufficiently answer these questions from the reviewers on this topic, and again, we point out that we are now in the process of making all raw data publically available for meta-analyses or additional hypotheses testing in the future using the included images and raw measurements. Copyright and legal permissions have now been given approval by the MPG, and these data will all be publically shared with the included DOI.

4) In the current Figure 5B the ratio for Dper reads 8.20, but in Figure 5D for the same species reads 8.28. I imagine this is just a typo, can they correct this prior to publication?

I believe this was a visual error based on the low resolution of the supplied figure (which actually reads “8.29”). However, in order to correct further any issues pertaining to decimal-related rounding of these average values, we have sought to include only two decimal places for each metric across all figures. We thank the reviewer for asking for clarity regarding this perceived discrepancy, and hope this is now resolved. The final figures should also be of high enough resolution that all text (and values) will be more legible as well.

5) Discussion:The authors should properly describe existing references in the Discussion. For example, the paper from Atkinson and Miller 1980 shows that in field experiment capture bait experiments, D. subobscura did not have a preference for baits located in a light vs dark areas. Meaning that field experiments so far do not support their hypothesis, this should be noted in the Discussion.

Agreed, and we have expanded this line/paragraph and section of the Discussion to better expand upon this already cited reference, namely pointing out that previous field trials in the 1980s have also attempted a similar analysis with field-collecting organisms, though we highlight that these field experiments should be repeated.

“Here, field studies in the 1980s using baited traps did not find a distinct difference in capture across light versus dark areas for *D. subobscura* (Atkinson and Miller, 1980). However, as mentioned before, additional fieldwork is still needed to continue to test our hypotheses outside of the laboratory, and within naturally occurring populations, especially in relation to abiotic measures of light gradients and GPS studies of canopy cover.”

6) Also in the Discussion, the statement should change from "changes in morphological investment can create dynamic differences in behaviour" to "changes in morphological investment correlate with dynamic differences in behaviour".

Thank you. We have changed “can create” into “correlate with”.

7) In the very extensive Discussion, they don't mention the alternative more plausible, hypothesis that behaviour might have switched first, through unknown neural basis, for some species to prefer shaded places (perhaps to avoid desiccation?) and thus become less visually guided. As the visual system is expensive to maintain this would produce a reduction in the visual system, the evolutionary pressure being energy preservation. Thus, reduction in visual investment being a consequence, not a driving force, of niche partitioning. This is indeed what happens with cave fish, where cave species have lost their eyes all together, but it is not thought that the fish went to the caves and speciated as a consequence of losing their eyes (as the authors seem to be suggesting in their discussion for flies), rather they specialised in the caves, and the no-longer useful eyes were lost. This seems a more parsimonious explanation for differential investment in vision and olfaction, than the proposed mechanism where reduced vision investment would drive niche partitioning. Not to mention that there is no evidence, nor circuit basis that could explain how larger eyes change phototaxis preference, which is probably computed in downstream circuits. Therefore, they should include in their Discussion the very possible hypothesis that flies reduced their visual investment as a consequence of niche partitioning, which occurred through yet unknown mechanisms.

We do not think these two hypotheses are necessarily contradictory, and instead, that they are perhaps more complementary. Niche partitioning can be driven from both ways at the same time, as an initial, albeit slight, preference for darkness or light, for example due to a spontaneous developmental mutation, can lead to a change in habitat preference and could then eventually solidify the new fly phenotype. However, this, basically, goes into one direction that remains puzzling across the animal kingdom. For example, nocturnal animals usually have two separate directions for visual investment, larger eyes or smaller eyes. This selection of the preferred sensory modality seems a bit weird across nocturnal animals, like owls (large eyes) vs. bats (small eyes). This evolutionary “decision” for sensory systems seems to rely on the respective starting point for the species or organism, if there already has been more visual or auditory investment, which is then perhaps more prioritized over time. These cavefish you mention of course have lost their eyes, as they are in absolute darkness, and therefore represent a very extreme example, although many deep-sea fish, in contrast, have actually increased their eye size investment (e.g. despite the complete darkness), as they perhaps have to observe all sorts of bioluminescence. Overall, this is a difficult concept to explain fully, especially without more data from additional animal species. Here again though, we highlight that to our knowledge, zero *Drosophila* species are nocturnal, which may prove an important developmental limitation to ponder as we continue to address the visual investment of these insects.

In the end, we concur with the original point from the reviewer, and have now sought to expand greatly a section or paragraph of the Discussion to include this alternative hypothesis within the lines pertaining to the potential order of evolutionary events. Moreover, we agree that this alternative interpretation cannot be ruled out, and we thank the reviewers for suggesting this expansion of the Discussion, which we hope will continue to create ongoing dialogue about the mechanisms for this observed shift in sensory systems between our examined species, and perhaps across vertebrate examples as well (i.e. cartilaginous fish).

“An alternative hypothesis would be that the phototaxis behaviors may have switched before morphology for some species, through an as of yet unknown mechanism, in order to push these species towards more shaded environments (i.e. perhaps to avoid desiccation pressures). […] As such, we continue to predict that niche partitioning, character displacement, and response to light gradients are the stronger initial driving forces of the evolution and speciation within the obscura group, where the novel environmental conditions drive sensory investment that in turn optimizes the courtship success of each species in this new niche.”

[Editors' note: further revisions were suggested prior to acceptance, as described below.]

There is just one comment that has not been fully addressed: the authors replied that they clarified the point that it remains to be shown whether the observed differences are truly interspecific but I could not read any statement regarding this fact in the Discussion. We suggest to add the following sentence (or similar) to the Discussion: "Please note that surface areas of the various head and thorax organs, as well as overall body size, were examined only for one strain per species. It thus remains to be confirmed that the observed differences are truly interspecific."

The sentence has been added to the Discussion and we have uploaded the new manuscript document.

Regarding the raw data supporting the findings of this study:– The raw data for Figure 3C, 4B is missing.

We have added the corresponding excel files for these figures as Supplementary files.

– The standard deviation for AL and OL of D. pseudoobscura was not calculated correctly based on the provided Excel file (see cells C45-F45 in Figure 2—source data 1).

We have corrected the error and replaced the corresponding excel file with the corrected one.

– It is really nice that the authors have made the data supporting the findings of this study available through EDMOND, the Open Access Data Repository of the Max Planck Society. However, not all their data is available on this platform. It would be great if the authors could also include all the videos analysed for this paper.

We are very sorry that we cannot upload all videos. Please see above.